# TimeSeriesExamAgent: Creating Time Series Reasoning Benchmarks at Scale

**Malgorzata Gwiazda**[1*†]**, Yifu Cai**[2*]**, Mononito Goswami**[2]**, Arjun Choudhry**[2]**,
Artur Dubrawski**[2]

[1]Technical University of Munich, Munich, Germany
`malgorzata.gwiazda@tum.de`

[2]Auton Lab, School of Computer Science, Carnegie Mellon University, Pittsburgh, US
`{yifuc, mgoswami, arjuncho, awd}@cs.cmu.edu`

## Abstract

Large Language Models (LLMs) have shown promising performance in time series modeling tasks, but do they truly understand time series data? While multiple benchmarks have been proposed to answer this fundamental question, most are manually curated and focus on narrow domains or specific skill sets. To address this limitation, we propose scalable methods for creating comprehensive time series reasoning benchmarks that combine the flexibility of templates with the creativity of LLM agents. We first develop `TimeSeriesExam`, a multiple-choice benchmark using synthetic time series to evaluate LLMs across five core reasoning categories: *pattern recognition*, *noise understanding*, *similarity analysis*, *anomaly detection*, and *causality*. Then, with `TimeSeriesExamAgent`, we scale our approach by automatically generating benchmarks from real-world datasets spanning healthcare, finance and weather domains. Through multi-dimensional quality evaluation, we demonstrate that our automatically generated benchmarks achieve diversity comparable to manually curated alternatives. However, our experiments reveal that LLM performance remains limited in both abstract time series reasoning and domain-specific applications, highlighting ongoing challenges in enabling effective time series understanding in these models. `TimeSeriesExamAgent` is available at `https://github.com/magwiazda/TimeSeriesExamAgent`

## 1 Introduction

Recent studies have successfully applied Large Language Models (LLMs) to time series analysis tasks including forecasting, anomaly detection, and classification (1; 13; 8; 19; 50; 51). These promising results raise a fundamental question: do LLMs possess genuine reasoning capabilities about the abstract concepts underlying time series data? Can they recognize trends, distinguish signal from noise, or understand causal relationships without relying on domain-specific shortcuts? Existing benchmarks designed to evaluate such capabilities face significant limitations– they are manually curated, expensive to extend, and typically focus on narrow domains or specific skills (44; 33). This creates a practical barrier for researchers and practitioners who need comprehensive evaluation tools but lack the resources to construct domain-specific benchmarks for their datasets.

To address this gap, we begin with a simple proof of concept, and introduce `TimeSeriesExam`, a controlled benchmark which uses *synthetic* time series to evaluate LLM reasoning across five core categories: *pattern recognition, noise understanding, similarity analysis, anomaly detection*, and *causality*. Initial results reveal two key insights: first, templated generation provides a viable mechanism for creating diverse and systematic evaluation questions; second, LLMs continue to struggle with abstract time series reasoning, even in these controlled settings.

---

* Equal contribution
† Corresponding author

Building on these insights, we tackle a broader practical challenge: how can we create benchmarks that reflect the domain-specific reasoning required in real applications, such as diagnosing arrhythmias from ECG signals or evaluating volatility regimes in financial markets? The key obstacle is that domain experts lack the time to manually construct comprehensive benchmarks, making expert-driven approaches impractical at scale. Inspired by recent advances in agent-based benchmark construction (6; 15), we address this challenge by combining our controlled synthetic benchmark with an extensible, agentic framework for automated domain-specific evaluation.

Experiments on multiple datasets spanning diverse domains reveal that (1) LLM performance varies substantially across domains, and (2) even state-of-the-art models struggle with complex reasoning tasks requiring integration of domain expertise and time series understanding.

Our contributions are as follows:

**Foundational benchmark.**  We introduce `TimeSeriesExam`, a controlled evaluation framework that systematically assesses whether LLMs understand core time series concepts. This templated approach provides valuable insights into LLMs' reasoning capabilities while enabling scalable question generation, though it remains limited to domain-agnostic skills on synthetic data. We provide the set of benchmark questions along with the accompanying code at `HuggingFace` and `GitHub`.

**Scalable framework.**  Building on this foundation, we propose `TimeSeriesExamAgent`, which combines the systematic nature of templates with the adaptability of LLM agents. Given any domain-specific dataset, `TimeSeriesExamAgent` automatically generates customized time series reasoning benchmarks at scale, integrating multi-perspective verification and optional human-in-the-loop refinement to ensure question quality and diversity.

**Multi-dimensional evaluation.**  We validate our approach across five datasets spanning four domains and demonstrate its effectiveness for domain-specific fine-tuning. Our automatically generated questions achieve diversity comparable to human-curated benchmarks.

## 2 RELATED WORK

**Synthetic Time Series Generation**  The generation of synthetic time series with controlled behaviors, such as trends and cyclic patterns, is fundamental for constructing scalable reasoning benchmarks. A common approach involves sampling from diverse random processes (18), such as Autoregressive Processes, which offer variability but lack control over specific patterns like cyclic behavior. To address this, (48) proposed a decomposition-based method, generating desired patterns by incorporating cyclic components into an additive model on top of random processes. More recent frameworks, such as (16) and (2), also leverage synthetic data generation for model training and evaluation. TabPFN constructs synthetic regression and classification tasks from random function priors, while Chronos employs large-scale transformer-based time series generation to capture realistic temporal dynamics. We build upon these studies, through the design of the TimeSeriesExam benchmark, by having a more diverse set of random processes and patterns, incorporating not only additive composition methods but also multiplicative and other forms of composition.

**Domain Specific Time Series Reasoning Benchmarks**  The task of creating domain-specific time series reasoning benchmarks is challenging. Current domain-specific benchmarks usually have limited scope and poor extensibility, since their curation often relies on templates or expert annotation. For instance, ECG-QA (33) and ECG-Expert-QA (44) focus on ECG interpretation, while EngineMT-QA (45) targets industrial settings. Automatic benchmark generation is a scalable alternative, but the quality and diversity of automatically generated questions is unclear. Without extensive verification, LLM-generated questions often require heavy manual curation (21; 25), which is both difficult and time-consuming, undermining the primary advantage of automation.

**Agents for benchmark creation**  LLM agents are autonomous systems which observe an environment, use LLMs to reason, and act towards achieving a well-defined goal. Recent work has shown the promise of using agents for creating benchmarks automatically. Most solutions adopt a

Table 1: Overview of time-series and multimodal datasets with curation and skill types (P—Prediction, R—Reasoning, PS—Practical skills). Prediction refers to supervised tasks such as forecasting or classification. Reasoning involves higher-level interpretation of time series signals (e.g., trend recognition). Practical skills extend reasoning into domain-specific contexts (e.g., classifying volatility regimes in finance). "+" represents that we can generate any number of samples, but we have already generated 3,000 of them.

| Title | Multi-domain | Curation | # Samples | Skill type | | |
|---|---|---|---|---|---|---|
| | | Fully Automatic | | P | R | PS |
| Time-MQA (21) | ✓ | ✓ | 200,000 | ✓ | ✓ | ✗ |
| Time-MMD (25) | ✓ | ✗ | 17,113 | ✓ | ✗ | ✗ |
| MT-Bench (9) | ✓ | ✗ | 22,000 | ✓ | ✓ | ✗ |
| ECG-QA (33) | ✗ | ✗ | 414,348 | ✓ | ✓ | ✓ |
| Context-is-key[1] (47) | ✓ | ✗ | 71 | ✓ | ✓ | ✗ |
| TimeSeriesExamAgent (ours) | ✓ | ✓ | 3000+ | ✓ | ✓ | ✓ |

multi-agent pipeline with planning, generation, validation, and evaluation modules (6). For example, (42) integrates exploratory evaluation using reinforcement learning, while (6) takes a description of a natural language task as input. However, most of these approaches are not tailored to time series and struggle to generate questions conditioned on numeric data. A recent solution incorporates time series, but is limited to a single-step design and lacks extensive verification (29).

**LLM-as-a-Judge Evaluation** LLM judges have enabled the research community to scale evaluations and to assess problems that are inherently difficult to capture with conventional metrics. However, LLM-as-a-judge also introduces challenges related to bias. We address this in two ways. First, we adopt methods such as G-Eval (26), which provide probabilistic results and thus improve both reliability and interpretability. Second, we account for intra-model bias, which is the tendency of a single LLM to exhibit systematic preferences, by drawing on the idea of panel-based evaluation (39). Specifically, we aggregate judgments from multiple LLMs, which reduces the influence of any individual model's bias and yields more stable scores.

## 3 TIMESERIESEXAM AND TIMESERIESEXAMAGENT

### 3.1 TIMESERIESEXAM: BUILDING SCALABLE BENCHMARK USING TEMPLATES

To begin, we investigate LLMs' understanding of fundamental time series concepts in a controlled experimental setting by introducing a manually curated, configurable benchmark that we call TimeSeriesExam.

To illustrate our approach, we present a proof-of-concept (PoC) showing how scalable benchmarks for evaluating LLMs' time series reasoning can be built from configurable templates. The aim is to demonstrate that template-based design enables systematic generation of diverse, controlled "exams" that probe specific reasoning skills. Our hypothesis is that once a small set of well-designed templates exists, new benchmark items can be generated automatically by varying parameters and contexts. To test this, we introduce TimeSeriesExam, a curated benchmark of fundamental time series tasks, in which we make two simplifying assumptions: (1) templates are created manually, and (2) evaluations are conducted in controlled synthetic settings where data properties are fully known.

**Composition.** TimeSeriesExam systematically assesses whether LLMs **understand** basic time series patterns such as trends and seasonality (*pattern recognition*), the concept of noise and other time series concepts in the presence of noise (*noise understanding*). It also evaluates LLMs on three different **reasoning** tasks: identifying abrupt deviation from "normal" behavior (12) (*anomaly detection*), comparing and contrasting statistical properties of 2 time series (*comparative reasoning*), reasoning about causality, specifically Granger Causality (14) (*causality*). As shown in Table 2, each category is further divided into sub-categories that represent more specific concepts within the broader category.

---

[1]The CiK benchmark contains 71 tasks, with the number of samples treated as a configurable hyperparameter.

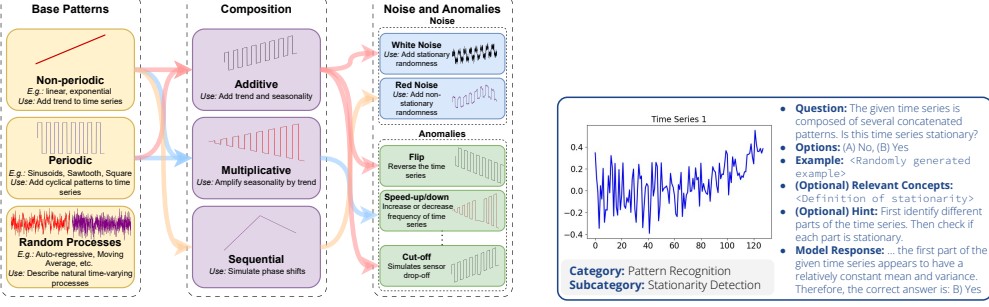

Figure 1: (*Left*) **Time Series Curation Pipeline:** The composition model generates controlled synthetic time series step-by-step. The pipeline enables diversity by combining different components to create numerous synthetic time series with varying properties. (*Right*) Each template evaluates a specific category, and includes a question, list of options, example question and answer pair for in-context learning, and optionally a hint and descriptions of complicated technical terms. Here, GPT-4o showcases its ability to transfer visual understanding and time series concepts into effective reasoning.

**Question Templates.** TimeSeriesExam comprises over 100 unique templates, carefully curated in collaboration with time series experts and cross-verified for accuracy, that can be used to generate any number of random questions. Each template (Fig. 1)(*Right*) evaluates a specific (sub-)category (e.g., *pattern recognition*), and comprises of a question (e.g., "Is this time series stationary?"), a list of options (e.g., "(A) Yes, (B) No"), and an example question and answer pair for in-context learning. Each template comes with a *hint* which breaks down complex questions into simpler steps and textual descriptions of complicated technical terms. By incorporating these relevant concepts, we can isolate an LLM's ability to understand time series concepts (e.g., whether the mean and variance remain constant) from its understanding of complex technical jargon (e.g., stationarity). Each option (e.g. "(A) Yes") is linked to a synthetic time series generator (Fig. 1)(*Left*) that produces a random time series as if the current option were true (e.g., a random stationary time series). This allows us to generate random but accurate time series at scale.

**Generating Questions.** We generate different questions from the same template by systematically varying the correct option and producing synthetic time series conditioned on the template and the correct option pair. Our simple and scalable approach, illustrated in Fig. 1(*Left*), involves sampling a small number of base patterns from a predefined pool and combining them using a composition function. Base patterns can be periodic (e.g., sine function), non-periodic (e.g., linear increasing

Table 2: Example template questions for different reasoning tasks. Each subcategory covers a specific aspect of time series understanding, guiding the model to reason about comparative, anomalies, and causal relationships.

| Category | Subcategory | Example question |
|---|---|---|
| Pattern Recognition | Trend | What is the most likely linear trend coefficient of the given time series? |
| | Cyclic | The given time series has sine wave pattern. How does its amplitude change from the beginning to the end? |
| | Stationarity | Is the given time series likely to be stationary after removing the cycle component? |
| | Regime Switching | Based on the given time series, how many different regimes are there? |
| | Statistical properties | Is the mean stable over time in the given time series? |
| | Random processes | Does the following time series exhibit a mean reversion property? |
| Noise Understanding | White Noise | Is the given time series a white noise process? |
| | Random Walk | Is the given time series likely to be a random walk process? |
| | Signal / Noise Ratio | You are given two time series with the same underlying pattern but different noise level. Which time series has higher magnitude of noise? |
| Anomaly Detection | | The following time series has two types of anomalies appearing at different time points. What are the likely types of these anomalies? |
| Comparative Analysis | Shape | Despite the noise, do the given two time series have similar patterns? |
| | Distributional | You are given two time series which are generated using a random walk. Are they likely to have the same variance? |
| Causality Analysis | Granger Causality | Is there Granger causality between the two time series? |

function), or random time-varying processes (e.g., AR process). Depending on the template's nature, the final step adds additive noise or anomalies using the anomaly injection process described in (12).

**Improving Questions Iteratively.** We use Item Response Theory (IRT) (27) to achieve finer grained control over the quality of randomly generated questions included in the `TimeSeriesExam`. During rounds of iterative refinement, question parameters are optimized to maximally distinguish the abilities of the candidate LLMs. Algorithm details are provided in App. A.2. The PoC establishes two takeaways: (i) template-based generation yields diverse, controllable items across core reasoning categories, and (ii) modern VLMs still struggle on higher-order time series reasoning, which motivates a scalable pipeline for real datasets with minimal input from an expert.

## 3.2 TIMESERIESEXAMAGENT: A SCALABLE DOMAIN-AGNOSTIC BENCHMARK CREATION TOOL

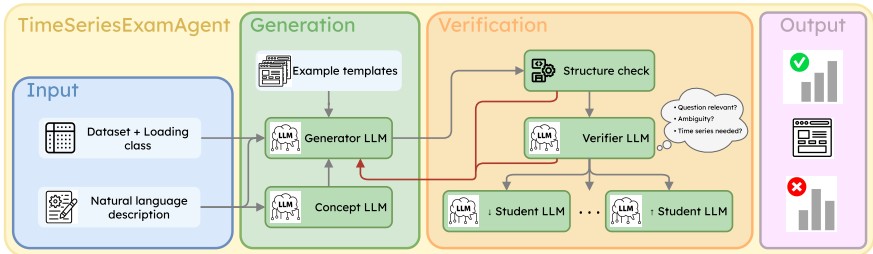

Figure 2: `TimeSeriesExamAgent` architecture. The user provides exam-making instructions and a custom dataset with minimal loading code. Agent outputs question templates – Python functions generated by a generator LLM and filtered through three progressive stages of verification (syntax and output format check, validation by LLM judge, capability-aligned filtering). Arrows denote data flow, red ones show direction for rejected templates.

Building on the PoC in the previous subsection that surfaced persistent gaps in LLMs' time series reasoning, we now focus on the practical need to evaluate models on domain-specific datasets at scale and with minimal expert effort. We propose `TimeSeriesExamAgent`, a multi-agent framework that combines planning, generation, and verification to enable automatic benchmark construction on user-provided datasets.

**Setup** An overview of the agentic framework is shown in Fig. 2. `TimeSeriesExamAgent` consists of two components that iteratively refine generation outputs: a Generation Agent and a Verification Agent. The Generation Agent takes as input a description of the natural language task $T$ and a dataset $D$. The description $T$ may include user guidelines for generation, contextual information about the dataset, or other relevant instructions. User provides a dataset class $D$ that supports basic operations such as querying the $i$-th sample. The Verification Agent employs a series of techniques to assess the robustness, relevance, and syntactic clarity of the generated questions.

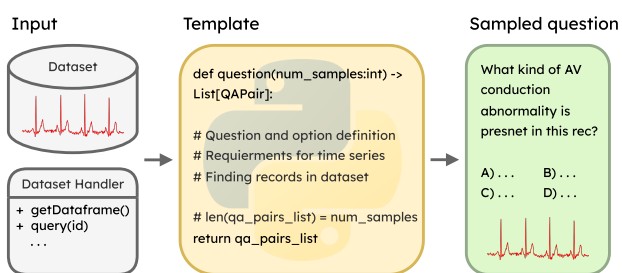

Figure 3: Given dataset information, `TimeSeriesExamAgent` generates question templates as Python functions that encode the question logic and support parameterized sampling of arbitrary instances.

**Generation** Motivated by `TimeSeriesExam`, we generate question templates instead of samples directly, as shown in Fig. 3, providing scalability and robustness to human errors. Given a dataset $D$ and task description $T$, a generator LLM produces a diverse set of templates conditioned on dataset

structure and domain-specific concepts. Templates define both the question format and the logic for selecting relevant time series instances and computing answers. Further details on prompt design, template structure, and generation workflow are provided in App. B.

**Verification**   We observe that LLM-based generation frequently produces errors or irrelevant outputs, motivating the need for a structured verification process. We propose a multistage verification and filtration process to check the accuracy and relevance of each template. If a template fails at any verification stage, it is returned to the generation agent with feedback. The generation is iterative with a maximum number of attempts, after which the ongoing template is permanently discarded to avoid excessive context length and cost from repeated failures.

**Structure check** We check whether the generated template can be executed successfully. This step isolates technical flows from content-based once, allowing for streamlined autonomous generation.

**Content verification** Certain aspects of quality control are particularly well-suited for using LLM-as-a-judge evaluation. We use an LLM verifier to assess the validity of each template. Details are in App. B.4.

**Capability-Aligned Filtering** Inspired by `TimeSeriesExam`, which leverages IRT to enhance differentiability across questions, we adopted a similar but localized framework that operates at the level of each template. To detect templates that generate overly simple or irrelevant exams, we evaluate them using a set of test-taking LLMs with varying capabilities. This approach is supported by the educational theory, particularly the expertise reversal effect (20). A template is permanently discarded if weaker LLMs achieve higher average accuracy than stronger models, as this typically indicates that the template is flawed or noisy rather than genuinely discriminative. A template is retained if performance scales with model capability, or if all models perform poorly, since such questions may still capture genuine difficulty. We provide hyperparameters in App. B.5 and other design specifics in App. B

## 4   TimeSeriesExamAgent– Experimental Setup, Results and Analysis

### 4.1   Iterative Refinement Enables Cost-Efficient Exam Generation

Most accepted templates pass verification within one or two refinement rounds, indicating that the generation prompts provide sufficient structure for stable template synthesis. Failed templates are discarded after a small number of attempts, preventing degeneration into unstable feedback loops and maintaining predictable behavior across datasets. The staged verification pipeline improves reliability by filtering execution errors and formatting issues before semantic judging, concentrating evaluation on viable candidates. The generation process yields diverse, executable, and domain-relevant questions without manual curation. The analysis of failure modes in App. F.2 shows that the remaining errors are systematic rather than random, which makes them diagnostic and correctable. Additional experiments on the sensitivity of the generation parameters are included in App. F.5.

The refinement loop also plays a role in reducing the costs of template creation. Under default configuration, the generation and verification process for one correct template costs approximately $0.09 in LLM API tokens.

### 4.2   State-of-the-art LLMs struggle on exams generated by TimeSeriesExamAgent

First, we generate a set of questions for each of the five real world datasets: PTB-XL (40), MIT-BIH (31), MIMIC-IV Waveform (30), yahoo finance stock dataset (37), and WeatherBench 2 (36). In total, we have 209 samples for YFinance, 197 samples for MIT-BIH, 151 samples for PTB-XL, 205 samples for MIMIC-IV Waveform, and 95 samples for WeatherBench 2. We sample 4 or 5 instances per template. Thus, the difference in the number of generated samples is a result of the template filtering mechanism above.

We select candidate models to cover a diverse range of performance levels, as indicated by the Open-VLM Leaderboard (11). In Table 3, we find that while general-purpose multimodal models such as

Table 3: Comparative performance of six vision–language models across medical (MIT-BIH, PTB-XL, MIMIC-IV Waveform (`MIMIC-IV W`)), financial (YFinance), and meteorological (WeatherBench 2) time series datasets. The results highlight dataset-specific strengths; nonetheless, all models achieve less than 55 mean accuracy, underscoring the difficulty of time series reasoning for current VLMs. The evaluation protocol is provided in App. C.3

| | Dataset | | | | | |
|---|---|---|---|---|---|---|
| Model | MIT-BIH | PTB-XL | MIMIC-IV W | YFinance | WeatherBench2 | Average |
| random guess | 0.25 | 0.25 | 0.25 | 0.25 | 0.25 | 0.25 |
| gpt-4o (17) | 0.416 | 0.424 | 0.385 | 0.586 | 0.389 | 0.440 |
| o3-mini (35) | 0.442 | 0.477 | 0.356 | 0.555 | 0.379 | 0.442 |
| Qwen2.5-VL-Instruct (3) | 0.411 | 0.490 | **0.439** | 0.572 | 0.368 | 0.456 |
| Gemma-3-27b-it (38) | 0.497 | **0.517** | 0.370 | 0.534 | 0.232 | 0.430 |
| GPT-5 | 0.533 | 0.450 | 0.424 | 0.617 | **0.547** | **0.515** |
| Gemini-2.5-Pro | **0.614** | 0.457 | 0.400 | **0.624** | 0.453 | 0.510 |

`GPT-5` perform well on weather-related questions, their performance is weaker on healthcare benchmarks. This contrast could suggest that the general reasoning ability does not always transfer across domains, particularly when tasks require domain-specific expertise or fine-grained interpretation of physiological signals. Secondly, We note that `GPT-5` strictly outperforms its predecessor, `GPT-4o`, across all categories. This improvement confirms that our benchmarks effectively distinguishes between model capabilities within the same family.

However, while the newer state-of-the-art models (`Gemini-2.5-Pro`, `GPT-5`) demonstrate improved reasoning capabilities, the gains are only incremental. Even the strongest model achieves only 51.5% average accuracy, which highlights the limitations of current LLMs. Excelling on domain-specific benchmarks likely requires multimodal pipelines with explicit tool-use and structured reasoning capabilities, such as agentic systems. In App. G, we highlight two main types of failure modes by studying responses from tested VLMs. **Perception**: As evidenced by our ablation on input resolution (DPI) or modality (text vs. vision), the best way to receive data depends on the specific question. App. F.4, F.3 provide further quantitative comparisons. **Compositional Reasoning**: Models do not fail on simple recognition, but on problems that require multi-step reasoning.

### 4.3 TIMESERIESEXAMAGENT GENERATES QUESTIONS WITH DIVERSITY COMPARABLE TO HUMAN-CURATED BENCHMARKS

We evaluate the diversity of questions generated by our framework against `ECG-QA` (33), a template-based benchmark built on `PTB-XL`. Our aim is to show that `TimeSeriesExamAgent` achieves comparable variety without manual template design. For each benchmark, we randomly sampled 50 questions and computed pairwise embedding distances. Embeddings were extracted using `Qwen/Qwen3-Embedding-8B`[2], the top open-source model on the Hugging Face MTEB leaderboard[3].

Table 4: Diversity of questions measured by embedding and normalized Levenshtein distances. Higher values indicate greater variability in phrasing.

| Benchmark Dataset | Mean ± Std | |
|---|---|---|
| | Embedding | Normalized Levenshtein |
| ECG-QA | $0.207 \pm 0.079$ | $0.519 \pm 0.157$ |
| TimeSeriesExamAgent (ours) | $\mathbf{0.301 \pm 0.070}$ | $\mathbf{0.542 \pm 0.039}$ |

As shown in Table 4, our framework achieves a level of diversity that is broadly comparable to human-curated benchmarks. This suggests that it can capture a range of question formulations without relying on handcrafted templates, which may help its scalability to other domains. For completeness, we include a visualization in App. C.2 to further illustrate this observation.

---

[2] https://huggingface.co/Qwen/Qwen3-Embedding-8B
[3] https://huggingface.co/spaces/mteb/leaderboard

Table 5: **Combined Quality Evaluation**. Scores (1–10) averaged across four criteria. Rows are grouped by their original source tables.

| Dataset | Mean Result | | | |
|---|---|---|---|---|
| | Specificity | Unambiguity | Domain Relevance | Answerability |
| *Finance Domain* | | | | |
| FinMME | 8.10 | **7.59** | 6.62 | 6.95 |
| MTBench | 6.88 | 6.11 | 8.35 | 7.29 |
| TimeSeriesExamAgent (ours) | **8.29** | 7.24 | **8.89** | **8.57** |
| *Medicine Domain* | | | | |
| ECG-QA | 5.60 | 5.77 | 8.17 | 8.47 |
| TimeSeriesExamAgent (ours) | **8.43** | **8.40** | **9.00** | **9.10** |

We employed an LLM-as-a-jury approach using G-Eval, where a panel of models (Gemini-1.5-Pro, GPT-3.5-Turbo, and Qwen2.5-VL-72B-Instruct) evaluated the quality of each question. To ensure cost efficiency, we selected relatively weaker models, as prior work shows this setup can maintain evaluation quality while mitigating intra-model bias (39). Each model independently assigned a score from 1 to 10 based on four criteria. The aggregated results, reported in Table 5, show that `TimeSeriesExamAgent` outperforms ECG-QA across all dimensions, particularly in specificity and answerability. This indicates that our framework generates precise, well-grounded, and domain-appropriate questions.

## 4.4 LLMs TRAINED ON OUR GENERATED SAMPLES EXHIBIT TRANSFERABLE REASONING SKILLS ON ESTABLISHED DATASETS

Another way to assess the value of `TimeSeriesExamAgent` is to test whether its generated data supports transfer learning. As a target model, we chose VLM `Qwen2.5-VL-3B-Instruct`. We first generated 2000 training samples using `TimeSeriesExamAgent` based on the `PTB-XL` dataset, while testing was conducted on 12000 randomly selected samples from the ECG-QA (34) test split of MIMIC-IV-based QA pairs. This ensured strict

Table 6: Results of VLM fine-tuning on the exams generated by `TimeSeriesExamAgent`. *General*: all incorrectly formatted responses are treated as wrong answers. *Parsable*: only correctly formatted responses are evaluated.

| Method | Accuracy | |
|---|---|---|
| | General | Parsable |
| Random answering | 34.9% | 34.9% |
| Base | 21.8% | 34.6% |
| Fine-tuned-confounded | 39.7% | 42.3% |
| Fine-tuned | **47.0%** | **49.7%** |

separation of data sources, exposing the model to different data within the same domain. Training parameters are provided in App. D. To isolate the effect of structural learning from actual gain in reasoning capability, in *Fine-tuned-counfounded* setup the LLM was trained on the questions from Finance and Weather domain instead of ECG related once. Ablation study on training data efficiency can be found in App. F.6.

Table 6 shows clear gains under the strict accuracy metric: the Base model achieves 21.8%, while fine-tuning on structurally similar but domain-irrelevant exam lifts accuracy to 39.7%. This confirms that the model benefits from instruction-following signals and structural regularities in the MCQ format. Training on `TimeSeriesExamAgent` –generated ECG exams lifts accuracy further to 47.0%, corresponding to a 216% relative improvement over the base model. This confirms the model also gained ECG reasoning capability. The fine-tuned model also surpasses the Random baseline (34.9%), indicating that agent-generated questions provide genuinely useful supervision rather than superficial patterning. Overall, these results suggest that synthetic, agent-curated exams can improve decision quality.

## 4.5 CHOICE OF LLM DOES NOT INTRODUCE BIAS FOR TIMESERIESEXAMAGENT

Although we only use the LLM-as-a-Jury system for linguistic properties check, we conducted experiment to confirm the consistency of juries with regard to choice of LLMs used, so that our

pipeline is not subject to bias from a specific set of LLM. We generate exams using default generator LLM `Claude-4-sonnet`, and evaluated using 3-model juries drawn from a fixed pool of LLMs (Gemini-2.0, Deepseek-V3.2, GPT-3.5 Turbo, Qwen-2.5-VL, LLama-3.3). We plotted inter-jury Pearson Correlation and Cohen's $\kappa$ in App. F.1 and observed that scores from most juries were moderately to highly correlated (Cohen's $\kappa \geq 0.5$). This confirms the consistency among different subset of LLMs and that our pipeline is not affected by bias arising from using a specific set of LLMs.

To confirm the choice of generator LLM do not introduce additional bias to the agentic pipeline, we fixed the jury to the default configuration (Gemini-2.0, GPT-3.5-Turbo, and Qwen2.5-VL-72B-Instruct), and compared two different generator LLMs: `Claude 4` and `DeepSeek V3.2`. We specifically picked DeepSeek because it is disjoint from both the Jury and Verifier model families (Qwen, Gemini, GPT), ensuring strict independence. Unlike the previous experiment, changing the generator alters the specific questions produced. Therefore, we fixed the underlying data source to MIT-BIH and evaluated the statistical distribution (mean $\pm$ standard deviation) of the jury scores across the generated artifacts. The results are presented in Table 9. Although DeepSeek V3.2 exhibits slightly higher raw means, the results are statistically comparable, with the scores of both models falling within one standard deviation of each other across all categories. This confirms that our generation pipeline is robust to the choice of the state-of-the-art LLM.

## 5 DISCUSSION, OPEN QUESTIONS AND OPPORTUNITIES

**Reliance on Expert-Generated Prompts**   A key limitation of `TimeSeriesExamAgent` is that the quality of the generated exams ultimately depends on the user instruction and coverage of the underlying time series dataset. For example, if important clinical instructions for the healthcare data set are absent, the resulting questions may not adequately capture the reasoning challenges faced in practice. In an offline sessions with cardiologists, we observed that when clinicians contributed targeted feedback during the prompt design stage, the resulting exams were consistently judged as more clinically valid and useful (See App. E). This highlights the importance of structured collaboration between automated systems and human experts, especially in high-stakes domains such as healthcare.

**Demand for human-in-the-loop evaluation.**   Building on the previous observation, we integrated optional human-in-the-loop modules into `TimeSeriesExamAgent` to facilitate a more practical deployment. These modules allow domain experts to refine templates, validate generated questions, and iteratively improve exam quality. Although we received encouraging anecdotal feedback from clinicians and practitioners who interacted with the system, the influence of such human feedback pipelines could not be systematically tested within the scope of this study. A formal evaluation of how human involvement impacts benchmark validity and downstream model assessment remains an important direction for future work and the community.

**Limited Evaluation Mode.**   In the current framework, questions are mainly evaluated by providing the time series as image input. In App. G.1, we provide a few case studies to highlight how input modality of time-series could impact model answers. These studies highlight the need for an intelligent decision-making tools, such as an agentic framework, to dynamically choose the most suitable representation. There is growing interest in agentic frameworks for time series analysis tasks (7; 49). Our benchmark provides a natural testbed for such systems, since many of the generated questions require multi-step reasoning, or direct computation over numeric data. Enabling agentic AI systems to autonomously write and execute code in order to answer our benchmark questions would provide valuable insights into their reasoning fidelity and robustness.

**Natural extension beyond time series.**   Although we focus on time series data, the underlying framework is not inherently restricted to this modality. Our design only assumes access to structured data and domain-specific prompts, making it extensible to other settings such as images, tables, or even multimodal combinations of signals and text. We chose time series as a starting point because it is a highly structured domain with well-established industrial applications.

# 6 CONCLUSION

This work first examined whether LLMs can reason about fundamental time-series concepts. To address these questions, we introduced `TimeSeriesExam`, a controlled benchmark for probing conceptual understanding, and `TimeSeriesExamAgent`, a scalable framework that enables practitioners to generate customized benchmarks from their own data. Our experiments show that while LLMs capture some surface-level patterns, they continue to struggle with more complex reasoning such as anomaly detection. At the same time, benchmarks generated by `TimeSeriesExamAgent` match or exceed the diversity and quality of human-curated datasets, and can even provide useful finetuning signals for downstream tasks. These results suggest that automated, agentic benchmark construction can help make evaluation more adaptive and domain-relevant.

## REPRODUCIBILITY STATEMENT

We release all evaluated datasets at `https://github.com/magwiazda/TimeSeriesExamAgent`. The evaluation protocol is described in App. C.3, and the appendix further provides implementation details of the pipeline, including prompts and configuration settings.

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

# A  TIMESERIESEXAM DETAILS

Table 7: TimeSeriesExam meta-information breakdown for each category. Each question is associated with a time series of length 128 time steps, and an example time series of length 64 time steps.

| Meta-information Type | Anomaly Detection | Similarity Analysis | Noise Understanding | Pattern Recognition | Causality Analysis |
|---|---|---|---|---|---|
| # Questions | 129 | 113 | 87 | 371 | 63 |
| %age questions with 2 time series | 31.01 | 100 | 14.94 | 3.77 | 100 |

## A.1  TIMESERIESEXAM TARGET MODEL EVALUATION

Table 8: Accuracy of model on each category of TimeSeriesExam.

| Model | Similarity | Pattern | Causality | Noise | Anomaly | Overall |
|---|---|---|---|---|---|---|
| Gemma-3-27B-IT(38) | 0.62 | 0.59 | 0.51 | 0.71 | 0.51 | 0.59 |
| GPT-4o(17) | **0.78** | **0.78** | **0.61** | **0.77** | 0.54 | **0.73** |
| Qwen2.5-VL-72B(3) | 0.60 | 0.45 | 0.49 | 0.40 | 0.22 | 0.44 |
| Gemini-2.5-Pro(10) | **0.78** | 0.76 | 0.57 | 0.65 | **0.59** | 0.71 |

Table 8 reports the accuracy on TimeSeriesExam in five categories of reasoning. We evaluated several state-of-the-art vision language models (VLMs) following the protocol described in App. C.3. Overall, GPT-4o achieves the strongest performance, closely followed by Gemini-2.5-Pro, while Qwen2.5-VL lags significantly behind. Across categories, models perform best on relatively shallow tasks such as similarity and pattern recognition, where surface-level cues often suffice. Performance drops sharply for more challenging categories. In particular, anomaly detection proves to be the most challenging one. This likely occurs because our anomaly detection questions are designed to go beyond easily identifiable visual irregularities (e.g., isolated spikes) and instead require compositional reasoning over temporal context, where subtle statistical deviations must be integrated with broader sequence behavior. As a result, success depends not only on perceptual recognition but also on higher-level reasoning about the data. By contrast, the causality subset is intentionally simplified to focus on Granger-style relationships with visually distinguishable patterns, reducing perceptual ambiguity and better isolating reasoning behavior. These results highlight that, while modern VLMs capture basic time series patterns, they fall short on higher-order reasoning tasks. We provide a case study in App. G.1

## A.2  ITERATIVE QUESTION IMPROVEMENT WITH IRT

IRT is a statistical framework that models the relationship between an individual's (or LLM's) latent trait (e.g., knowledge, ability) and their responses to a set of items (e.g., questions on a test). It is a valuable tool in exam development as it helps to identify weak exam items, ensures consistent scoring across different versions of the exam, and also allows tailoring the testing experience to the LLM's abilities.

Our primary objective is to design a TimeSeriesExam where each question can maximally distinguish the abilities of the candidate LLMs. We use the two-parameter logistic (2PL) model for this. Formally, for LLM $j$ with ability $\theta_j$, and question $i$ with difficulty $b_i$, discrimination ability $a_i$, the 2PL model defines the probability of a correct response as:

$$\mathbb{P}(r_{ij} = 1 \mid a_i, b_i, \theta_j) = \frac{1}{1 + e^{-a_i(\theta_j - b_i)}} \tag{1}$$

Each exam typically undergoes 1–3 rounds of iterative refinement. In each round, all candidate models take the exam. Based on their responses, we fit the parameters of Equation 1 using maximum likelihood estimation (MLE). Then, we drop $X\%$ of samples with the lowest sum of difficulty and discrimination ability. Finally, we randomly re-generate questions from the dropped templates. This iterative process is detailed in Algorithm 1 and the hyper-parameters of the fitting process are provided in App. A.3. We demonstrate the increase in the differentiability parameter across the iteration rounds and provide an analysis of the trajectory in App. A.4

---

**Algorithm 1** Iterative Dataset Refinement with IRT and Resampling

---

**Require:** num_iterations = 3, drop_percentage = 0.2, initial dataset $D_0$
 1: $D \leftarrow D_0$
 2: **for** iteration = 1 to num_iterations **do**
 3:     **Evaluate** each candidate $i$ on $D$, and obtain the response set $R = \{r_{ij} \mid r_{ij} = 1$ if candidate $i$ correctly answers question $j\}$
 4:     **Fit** the IRT model to obtain the discrimination parameters $\mathbf{A} = \{a_j \mid j \in \text{Questions}\}$ and difficiulty parameter $\mathbf{B} = \{b_j \mid j \in \text{Questions}\}$
 5:     **Normalize** set $\mathbf{A}$ and $\mathbf{B}$ between 0 and 1, and calculate score $\mathbf{S} = \{b_j + a_j \mid j \in \text{Questions}\}$
 6:     **Find** $\mathbf{S}'$ which is the score for samples that are answered correctly by the best model in the round
 7:     **Find** the index set $I = \{j \mid a_j < \text{Quantile}(\mathbf{S}', \text{drop\_percentage})\}$, where $a_j$ is less than the drop_percentage quantile of $\mathbf{A}$
 8:     **for** each $j \in I$ **do**
 9:         **Resample** a new question $q'$ from the same category as question $j$
10:         Set $D[j] \leftarrow q'$
11:     **end for**
12: **end for**
13: **return** $D$

---

## A.3 IRT MODEL PARAMETERS

The IRT models are fitted using library *py-irt* (22). The parameters are epochs=2000, lr=0.1, lrdecay=0.9999, dropout=0.5, hidden=100

## A.4 AVERAGE SAMPLE DISCRIMINATION PARAMETER OVER ROUNDS

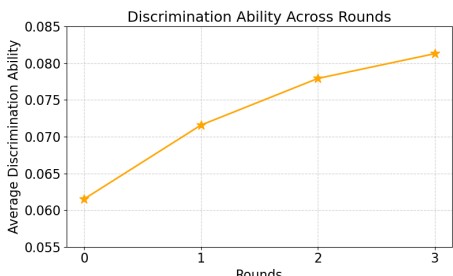

Figure 4: The sample average discrimination parameter across rounds shows an upward trend, indicating an improved ability of the questions to differentiate candidates with varying levels of ability.

## A.5 DROPPED DATASET DISTRIBUTION PER ROUND

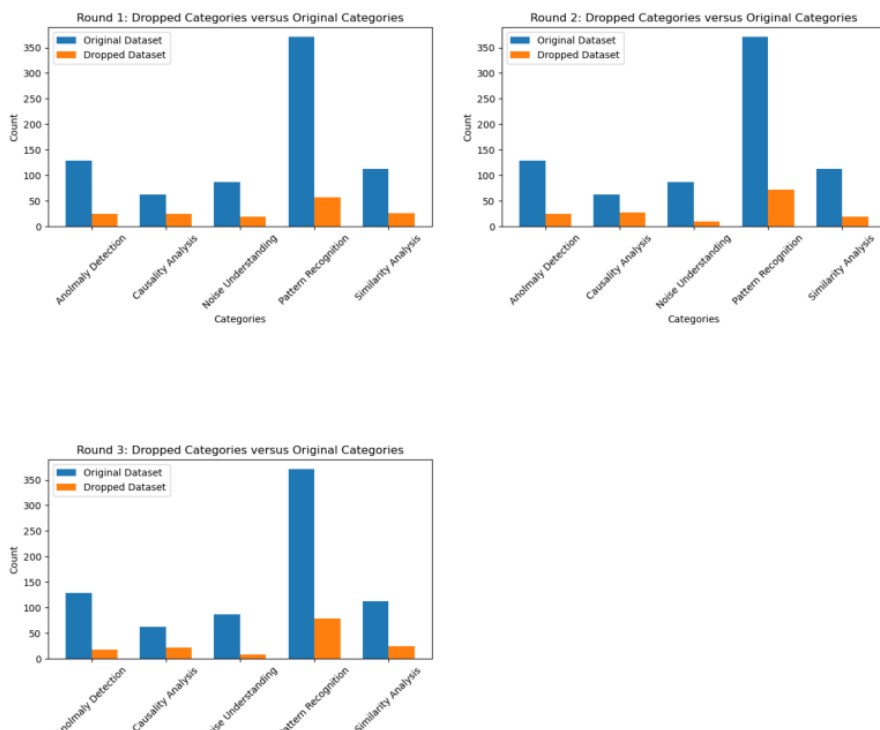

Figure 5: Dropped Dataset Distribution per round. Dropped category distribution per round generally mirrors the overall category distribution.

We can observe in Fig. 5 that the proportion of dropped questions for each category is approximately uniform. The difference in number of questions in each category is a result of different template curated.

# B   TIMESERIESEXAMAGENT DETAILS

## B.1   GENERATION WORKFLOW

We rely on two stages of generation for the templates: planning and generating, inspired by the chain-of-thought (CoT) prompting(46).

**Generation planning**   To provide a relevant and diverse set of templates, we rely on a comprehensive list of domain-specific concepts. There are several ways our pipeline generates a list of concepts:

1. LLM generation: User guidelines and dataset descriptions are provided as input to an LLM, which proposes the concepts.
2. Web Search: We provide the option for generator LLM obtain concepts through web search.
3. Retrieval Augmented Generation: As an option, the user could also provide a relevant file from which the LLM reads and generates concepts(23).

**Template generation**   As input to our generator, the following components are provided:

- User-provided guidelines: a document containing the user's goal or specific requirements,
- Dataset description: a list of columns and example values with ranges from the dataset, with a short usage example,
- List of concepts: generated in previous step. For each template, our pipeline will choose a concept at random to ensure diversity.
- Example templates: few-shot examples presenting required structural elements (5). These templates are constant and domain-agnostic, as they describe only the format and structure of the correct template. Their content is therefore not relevant.

### B.1.1   GENERATION PROMPT

```
Here is the goal of the exam questions:
{user_info_text}

Here are sample concepts on which you can base your question generation:
{concept_conversation}

Use the concept numbered {concept_no} from the list to guide the design of your question
    template.

Here is the description of the dataset you will use to generate the question:
{dataset_describe}

In your template, use the provided 'user_dataset' object. Use its 'query(index)' method to
    load relevant time series data.

Do not select time series randomly. First, formulate the question, and then choose a time
    series that fits its logic and reasoning needs.

Generate one function-based question template now.
```

### B.1.2   TEMPLATE CONSTRUCTION

A template is implemented as a Python function. It contains a formatted string for the question and options, together with parameters that control how many questions to generate. For each question, the template samples a pair $(x_i, z_i)$ from the dataset $D$, where $x_i \in \mathbb{R}^{n \times d}$ is a time series with $n$ observations and $d$ variables, and $z_i$ is an auxiliary array containing metadata or labels related to the series. Later, a rule-based calculation is performed to determine the correct answer from the time series. For example, in a trend-detection template, the function computes the linear trend coefficient of $x_i$ and selects "Yes, there is a linear trend" if the coefficient exceeds a specified threshold. In addition to such signal-derived logic, templates can also utilize the auxiliary property $z_i$, effectively transforming classification problems into question–answer form. For instance, if an ECG series in the dataset is labeled as exhibiting atrial fibrillation, the template can present this label as one of the multiple-choice options. Each generated sample consists of the question, its options, the correct answer, and one or more associated time series represented as numerical values.

## B.2 Supporting Mechanisms for Generation

**Detractors** In addition, the mechanism of plausible but incorrect answer choices was implemented. The LLM is prompted to reflect on possible mistakes that the test taker might make while solving the exam. Using this knowledge, misleading, incorrect option choices can be generated.

**Context Condensation** A common issue we encountered in the framework was context window overflow during exam regeneration. To mitigate this, we applied context condensation, which reduces the number of tokens while preserving essential information. In our setup, the agent generates templates in a conversational manner. The process begins with a generation prompt, followed by a message containing the generated exam. If errors occur or the exam is rejected during verification, the feedback and regenerated exams are appended to the conversation. Several context condensation techniques exist, such as windowing (4) and context compression (32). We adopt a summarization-based method (41; 43), which has shown strong results in prior work and fits our use case. Specifically, we summarize non-recent pairs of failing exams and error messages into short descriptions that highlight the issues encountered. These summaries provide the LLM with concise feedback, supporting the generation of higher-quality templates.

**RAG/web search** In our setup, LLMs can also make use of external knowledge sources. The agent has two options: (i) a Retrieval-Augmented Generation (RAG) tool (24), which pulls information from a structured corpus such as a PDF with domain materials, and (ii) web search, which provides access to more up-to-date or niche information. The retrieved content is then used to support concept generation, helping the model produce more accurate and comprehensive outputs.

## B.3 Mitigating Circularity in LLM-Based Validation and Evaluation

To mitigate the risk of circularity when using LLMs to evaluate artifacts generated by other LLMs, we incorporated strict safeguards into both the generation methodology and the subsequent evaluation process.

**LLM Verifier** In our framework, the verifier is intentionally restricted to shallow linguistic and structural checks (e.g., domain relevance and ambiguity detection) rather than deep correctness verification, reducing the risk of propagating model-specific reasoning biases.

**Capability-Aligned Filtering** Our approach does not introduce bias caused by test-taking LLMs' performance – only the occurrence of an inverted performance gap results in template rejection. This filtering mechanism does not affect the performance or ranking of other models.

Additionally, we use disjoint model families for crucial roles in `TimeSeriesExamAgent` to avoid same-family optimization effects and to better approximate an external auditing setup.

## B.4 LLM Verifier Specifics

For each template, we use an LLM to evaluate the generated question. Specifically, we ask:

- Is the question relevant to the given concept?
- Does answering the question require the provided time series?
- Are the question and answer free from ambiguity and bias?

We use a binary scheme: a template must pass all categories to be accepted. Failure in any of the categories above triggers regeneration, ensuring robustness.

### B.4.1 Validation Prompt

```
You are an expert validator of question templates involving reasoning over
{exam_type} time series data.
You are given an exam question template:

{exam_template}
```

```
Your task is to validate the question template using the following criteria:
1. Is the question relevant to {exam_type} time series analysis?
2. Would you need the time series itself to answer the question?
3. Are there no ambiguity in the question or its answer?

If the answer to all is YES or MOSTLY YES, return only the number 1.
If the answer to either is NO, return your objections.
Return 1 (do not include any additional text then) or describe your objections.
```

### B.5 FRAMEWORK HYPERPARAMETERS

In this section, we list all the hyperparameter used for our agentic workflow.

1. Generator LLM: the LLM used to generate concepts and the corresponding template. We used claude-sonnet-4-20250514 (initial generation with `reasoning_effort="medium"`). As a result, models developed by Anthropic are excluded from subsequent evaluations.

2. Concept LLM: the LLM used to generate concepts. We used gpt-4o-2024-08-06.

3. Structure verification: Each template is sampled $k = 3$ time to create $k$ exam questions. Potential syntax errors, incorrect dataset querying, and bad output structure is detected, ensuring compatibility with the rest of the pipeline.

4. Verifier LLM: the LLM used to verify templates. We used gpt-4o-2024-08-06.

5. Student LLMs: the student LLMs we use to check the exam differentiability. Currently we have two student LLMs: stronger: gpt-4o-2024-08-06 and weaker: gpt-4o-mini. The strength of a model is determined based on the OpenVLM leaderboard. Moreover, family of student models was unified to minimize architectural variations while providing wide enough capability gap. For each template under evaluation, students receive the same set of 3 samples to answer.

6. Exam type: We are generating the data connected to specific domain. We used "ecg", "medicine", "finance", "weather" and "mechanical".

7. Few-shot examples: 9 templates prepared beforehand were used to present the desired structure to the generator LLM. For each generation, 3 templates were randomly selected and included in the prompt as few-shot examples. This introduces variability into the generation process, enhancing diversity.

8. Regeneration patience: Templates requiring multiple regeneration cycles were generally of lower quality. In our experiments, we set a maximum of 3 regeneration attempts.

### B.6 EXAMPLE OF NATURAL LANGUAGE DESCRIPTION

```
I want to create time series exam testing model understanding of finance time series data.

To load the data, use the provided '''user_dataset''' object.

Given time series come from Yahoo Finance, include closing price of a stock. Interval between
    samples is 1 day.
Make sure that the length of time series (total number of samples of one or two time series)
    does not excide 3000.

Please make sure that exams cannot be answer without timeseries!
```

## B.7 EXAMPLE OF QUESTION TEMPLATE

```python
def question_6(num_samples, verbose=False):
    hyperparameters = {
        "min_trend_days": 20,
        "max_series_length": 3000,
        "trend_strength_threshold": 0.7,
        "momentum_window": 10,
    }

    question = "Analyzing the price movements of {ticker} over the given time period, does the
     price trend demonstrate strong momentum and sustainability, or does it show signs of
    weakness and potential reversal?"

    options = [
        "The trend shows strong momentum with consistent directional movement and minimal
     pullbacks, suggesting the trend is likely to continue.",
        "The trend shows signs of weakness with frequent reversals and inconsistent momentum,
     suggesting a potential trend change.",
        "The trend shows mixed signals with alternating periods of strength and weakness,
     making direction unclear.",
        "The price movement shows no clear trend pattern, indicating a ranging or sideways
     market."
    ]

    def calculate_trend_strength(prices):
        if len(prices) < hyperparameters["min_trend_days"]:
            return None, None

        returns = np.diff(prices) / prices[:-1]

        # Calculate momentum consistency
        positive_days = np.sum(returns > 0)
        negative_days = np.sum(returns < 0)
        total_days = len(returns)

        directional_consistency = max(positive_days, negative_days) / total_days

        # Calculate average magnitude of moves
        avg_abs_return = np.mean(np.abs(returns))

        # Calculate trend persistence (consecutive moves in same direction)
        consecutive_moves = []
        current_streak = 1
        for i in range(1, len(returns)):
            if np.sign(returns[i]) == np.sign(returns[i-1]):
                current_streak += 1
            else:
                consecutive_moves.append(current_streak)
                current_streak = 1
        consecutive_moves.append(current_streak)

        avg_streak = np.mean(consecutive_moves)
        max_streak = max(consecutive_moves)

        # Determine overall trend direction
        overall_return = (prices[-1] - prices[0]) / prices[0]
        trend_direction = "up" if overall_return > 0 else "down"

        return {
            "directional_consistency": directional_consistency,
            "avg_abs_return": avg_abs_return,
            "avg_streak": avg_streak,
            "max_streak": max_streak,
            "overall_return": abs(overall_return),
            "trend_direction": trend_direction
        }, returns

    qa_pairs = []
    df = user_dataset.get_dataframe()

    attempted_tickers = set()

    while len(qa_pairs) < num_samples:
        if verbose:
            print(f"[Question 6] Generating question {len(qa_pairs)} / {num_samples}")

        # Select a ticker that hasn't been attempted
        available_tickers = [i for i in df.index if i not in attempted_tickers]
        if not available_tickers:
            break
```

```
    ticker_id = random.choice(available_tickers)
    attempted_tickers.add(ticker_id)

    ticker = df.loc[ticker_id, 'ticker']
    prices = user_dataset.query(ticker_id)

    if len(prices) < hyperparameters["min_trend_days"]:
        continue

    # Limit series length
    if len(prices) > hyperparameters["max_series_length"]:
        start_idx = random.randint(0, len(prices) - hyperparameters["max_series_length"])
        prices = prices[start_idx:start_idx + hyperparameters["max_series_length"]]

    # Select a subset for analysis (to make question more focused)
    analysis_length = min(len(prices), random.randint(50, 200))
    start_idx = random.randint(0, len(prices) - analysis_length)
    analysis_prices = prices[start_idx:start_idx + analysis_length]

    trend_metrics, returns = calculate_trend_strength(analysis_prices)
    if trend_metrics is None:
        continue

    # Determine answer based on trend strength metrics
    strength_score = (
        trend_metrics["directional_consistency"] * 0.4 +
        min(trend_metrics["avg_streak"] / 5, 1.0) * 0.3 +
        min(trend_metrics["overall_return"] * 10, 1.0) * 0.3
    )

    if strength_score >= hyperparameters["trend_strength_threshold"] and trend_metrics["
max_streak"] >= 5:
        answer = options[0]
    elif strength_score < 0.4 or trend_metrics["directional_consistency"] < 0.6:
        answer = options[1]
    elif 0.4 <= strength_score < hyperparameters["trend_strength_threshold"]:
        answer = options[2]
    else:
        answer = options[3]

    question_text = question.format(ticker=ticker)

    qa_pairs.append({
        "question": question_text,
        "options": options,
        "answer": answer,
        "ticker": ticker,
        "ts": analysis_prices,
        "relevant_concepts": ["Volume-Price Trend Correlation", "Trend Strength Analysis",
 "Price Momentum"],
        "domain": "finance",
        "detractor_types": ["Incorrect trend interpretation", "Misunderstanding momentum
signals"],
        "question_type": "multiple_choice",
        "format_hint": "Please answer the question and provide the correct option letter,
e.g., [A], [B], [C], [D], and option content at the end of your answer. All information
need to answer the question is given. If you are unsure, please provide your best guess.",

    })

return qa_pairs
```

## B.8 EXAMPLES OF GENERATED QUESTIONS

**ECG Question Example**

**Q:** Analyze the P-wave morphology and amplitude characteristics in this recording. What atrial abnormality is present?

A. RAO/RAE: Right atrial overload/enlargement with prominent P-waves

B. LAO/LAE: Left atrial overload/enlargement with bifid P-waves

C. Normal P-wave morphology with no atrial abnormalities

D. Absent P-waves indicating atrial fibrillation

**answer:** LAO/LAE: Left atrial overload/enlargement with bifid P-waves

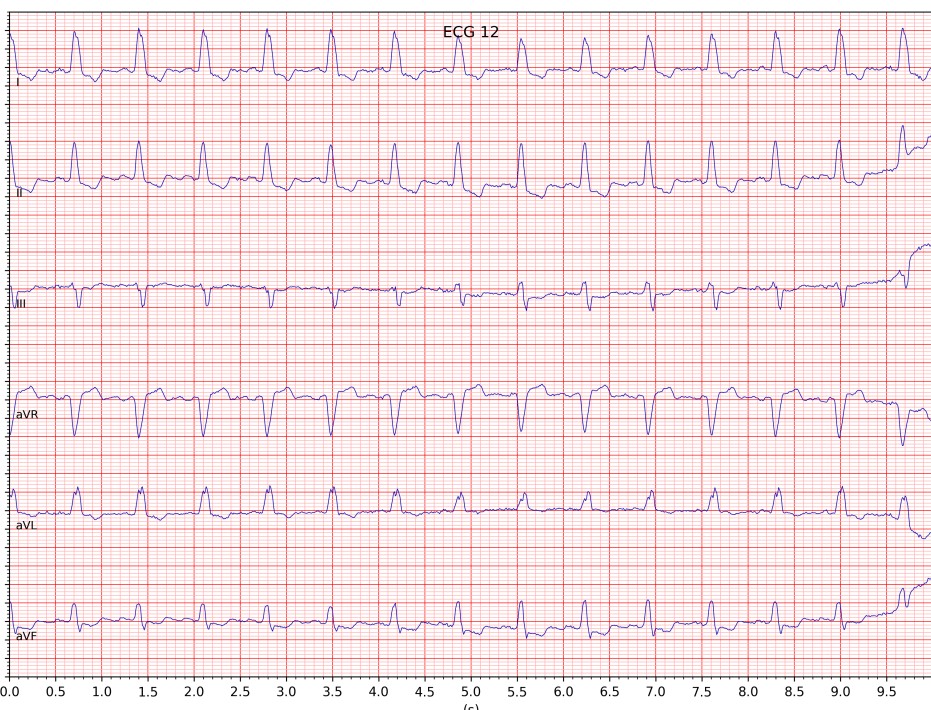

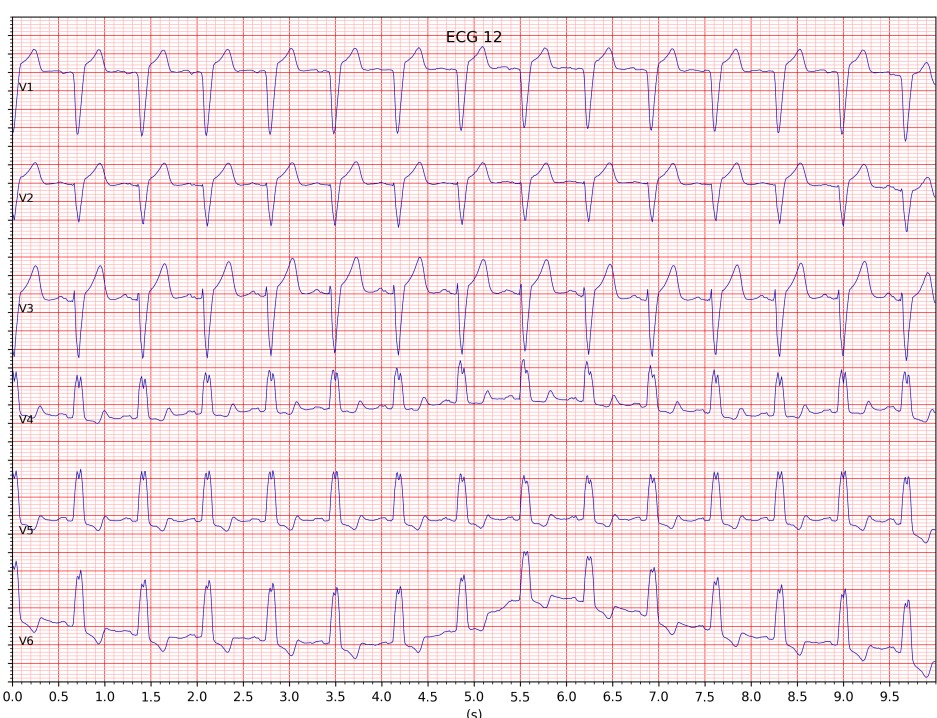

**Finance Question Example**

**Q:** Based on the daily closing price data for MAA over the past 2000 trading days, what does the Relative Strength Index (RSI) analysis reveal about the stock's momentum condition at the end of the period?

A. The stock is in overbought territory with RSI above 70, suggesting potential selling pressure.

B. The stock is in oversold territory with RSI below 30, suggesting potential buying opportunity.

C. The stock shows neutral momentum with RSI around 50, indicating balanced buying and selling pressure.

D. The stock shows strong upward momentum with RSI consistently increasing but not yet overbought.

**answer:** The stock shows neutral momentum with RSI around 50, indicating balanced buying and selling pressure.

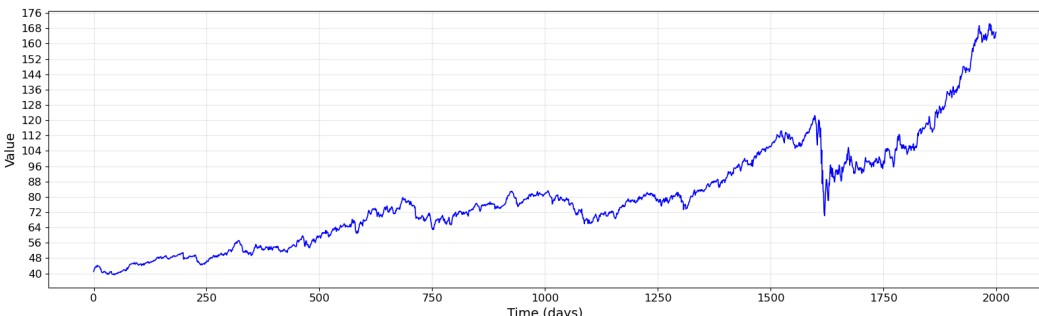

# C EVALUATION DETAILS OF GENERATED RESULT FROM TIMESERIESEXAMAGENT

## C.1 LLM-AS-A-JURY

We evaluated a set of generated questions using the LLM-as-a-jury approach. Below are example criteria we applied for ECG evaluation:

```
1. SPECIFICITY
You are an expert judge evaluating the specificity of ECG multiple-choice questions.
The questions normally come together with relevant time series data, which should be analized
     to answer the question correctly. It is not includded in currently evaluated samples.

Evaluate the specificity of the generated ECG multiple-choice question.

A good question should target a single phenomenon.

Evaluation steps:
1. Read the question and all answer options.
2. Determine if the question targets a single, clearly defined ECG finding or clinical
     interpretation.
3. Assess the ratio of unique medical terms to general words.
4. Penalize if:
   - The question is overly broad or open-ended (e.g., "Is this ECG normal?").
   - The wording leaves diagnostic interpretation unclear.
   - The question covers multiple unrelated phenomena.

Score highest if the question has one precise focus (e.g., "Is there ST elevation in lead V3
     ?").

Score from 1-10 where:
- 10: Excellent specificity with clear, focused medical terminology targeting a single
     phenomenon
- 7-9: Good specificity but could be more focused
- 4-6: Moderate specificity with some clarity issues
- 1-3: Poor specificity, too broad, or covers multiple unrelated phenomena

Respond with just a number from 1 to 10, followed by a brief explanation for your score.

2. UNAMBIGUITY
You are an expert judge evaluating the unambiguity of ECG multiple-choice questions.
The questions normally come together with relevant time series data, which should be analized
     to answer the question correctly. It is not included in currently evaluated samples.
Task: Evaluate if the question and answers can be objectively assessed without multiple
     interpretations.

Evaluation steps:
1. Read the question and all answer options.
2. Determine if the question can be objectively assessed.
3. Check if the answers are clear and unambiguous.
4. Penalize if:
   - The question uses subjective terms (e.g., "Does this look strange?").
   - The answers are open to multiple interpretations.
   - The question cannot be objectively answered.

A good question should be clear and objective (e.g., "Is there tachycardia?").

Score from 1-10 where:
- 10: Completely unambiguous and objective with crystal clear question and answers
- 7-9: Mostly clear with only minor ambiguities
- 4-6: Moderately clear but has some ambiguous elements
- 1-3: Highly ambiguous, subjective, or open to multiple interpretations

Respond with just a number from 1 to 10, followed by a brief explanation for your score.

3. DOMAIN RELEVANCE
You are an expert judge evaluating the domain relevance of ECG multiple-choice questions.
The questions normally come together with relevant time series data, which should be analized
     to answer the question correctly. It is not includded in currently evaluated samples.
Task: Evaluate if the question actually pertains to ECGs and medicine.

Evaluation criteria:
1. Does the question contain medical and ECG-specific terminology?
2. Is the question relevant to ECG interpretation and medical diagnosis?
3. Is the question related to ECG interpretation?
4. Does the question have proper medical context?
```

```
A good question should contain relevant medical terms (e.g., "QRS," "arrhythmia," "P wave")
    and pertain to ECG interpretation.

Score from 1-10 where:
- 10: Highly relevant to ECG domain with extensive proper medical terminology
- 7-9: Good domain relevance with appropriate medical terms
- 4-6: Moderate relevance but could be more medically specific
- 1-3: Poor medical relevance or contains primarily non-medical terms

Respond with just a number from 1 to 10, followed by a brief explanation for your score.

3. ANSWERABILITY
You are an expert judge evaluating the answerability of ECG multiple-choice questions.
The questions normally come together with relevant time series data, which should be analized
    to answer the question correctly. It is not included in currently evaluated samples.
Task: Evaluate if the question can be answered based on ECG data analysis.

Evaluation steps:
1. Read the question and all answer options.
2. Determine if the question can be answered by analyzing ECG waveform data.
3. Assess whether the question requires time series analysis or could be answered without it.
4. Penalize if:
    - The question asks about non-ECG factors (e.g., "Was the patient nervous?").
    - The question can be answered without analyzing the ECG time series data.
    - The question is too general and doesn't require specific ECG analysis.

Score highest if the question requires specific ECG time series analysis (e.g., "Is there
    atrial fibrillation?").
Give fewer points if the question can be answered without time series data.

Score from 1-10 where:
- 10: Requires specific, detailed ECG analysis and is fully answerable from the data
- 7-9: Mostly answerable from ECG data but could be more specific
- 4-6: Partially answerable from ECG but has some limitations
- 1-3: Cannot be answered from ECG data or is too general/unrelated

Respond with just a number from 1 to 10, followed by a brief explanation for your score.
```

## C.2 T-SNE EMBEDDING PLOTS

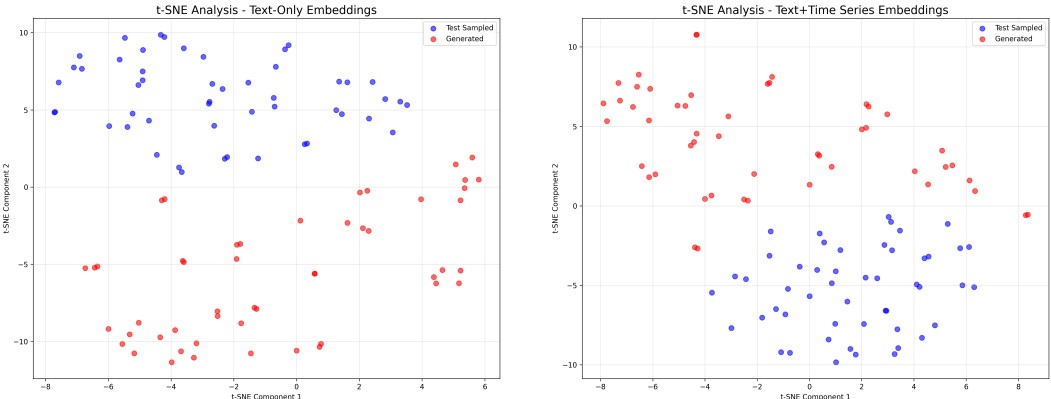

Figure 6: t-SNE analysis of embeddings: (left) text-only vs. (right) text and time series concatenated together.

To visualize distributional differences, we applied t-SNE (28), which preserves local distances between samples. As shown in Fig. 6, questions generated by our framework form a more widely scattered distribution, confirming the higher diversity observed in Table 4.

We ran the t-SNE algorithm using the scikit-learn implementation with the random seed fixed to 42, in order to ensure full reproducibility of the dimensionality reduction results across different runs. The other parameters were set to default.

The text embeddings were generated using the SentenceTransformer model (Qwen3-Embedding-8B), while the time-series embeddings were obtained from MOMENT-1-large and averaged across

leads or multiple time series when applicable. The two vectors were then combined through direct concatenation to form a joint embedding.

## C.3 QA Samples Evaluation Protocol

All used models were accessed by API with `LiteLLM` Python library. The following API providers were used with default parameters:

- Closed source models – OpenAI API, Anthropic API
- Open source models for `TimeSeriesExamAgent` generated exams – Hugging Face Inference Providers API
- Open source models for `TimeSeriesExam`– Novita AI [4]

During the evaluation, the images of the plots were encoded with base64 encoding and provided to the models. Plots were created with DPI = 50. We used setup without context condensation.

---

[4]`https://novita.ai/`

# D   TIMESERIESEXAMAGENT-BASED FINE-TUNING PARAMETERS

| Hyperparameter | Value |
|---|---|
| Base model | Qwen2.5-VL-3B-Instruct |
| GPU setupe | 4*NVIDIA RTX A6000 48GB GPU |
| Frameworks | Hugging Face Accelerate, DeepSpeed ZeRO 3 stage |
| Train samples | 2000 |
| Warm-up steps | 16 |
| Batch size per device | 1 |
| Gradient accumulation steps | 8 |
| Learning rate | 5e-5 |
| Optimizer | AdamW |
| Learning rate scheduler | Cosine |
| Weight decay | 0.1 |
| LoRA rank (r) | 16 |
| LoRA alpha | 16 |
| LoRA dropout | 0.0 |
| LoRA target modules | q_proj, k_proj, v_proj, o_proj, gate_proj, up_proj, down_proj |

# E  FEEDBACK IMPACT

One of the challenges we observed during question generation was the misuse of domain-specific jargon. Although the generated questions were grammatically correct, they sometimes included terminology that did not align with standard ECG practice. This can lead to confusion for clinicians, as non-standard phrasing undermines clarity and clinical relevance.

The following generated question contains terminology that was later identified as suboptimal:

> **Q:** Examine this Lead II ECG recording and measure the QRS voltage amplitudes throughout the tracing. Based on the peak-to-peak QRS amplitudes observed, what voltage abnormality is present?

The clinicians noted that certain expressions in the question do not reflect standard ECG terminology. In particular, the phrase *"peak-to-peak QRS"* was considered inappropriate. To address this, the natural language description was refined by adding the following instruction:

```
Please frame your questions in a way that is clear and natural for
ECG specialists (i.e., adjust terminology accordingly).
```

Following this modification, a second round of consultation confirmed that the issue of non-standard jargon had been resolved. An example of an improved question generated with the revised prompt is shown below:

> **Q:** Based on QRS voltage amplitude measurements across all 12 leads in this ECG, which ventricular condition is most likely present?

# F Ablations and Additional Analyses

## F.1 LLM-as-a-Jury score consistency analysis

To evaluate the consistency of jury-based scoring, we begin by selecting a diverse set of open- and closed-source models (Gemini-2.0, DeepSeek-V3.2, GPT-3.5 Turbo, Qwen-2.5-VL, and Llama-3.3). We form all possible triplets from this set and compute jury scores using the same procedure as in Table 5. For each triplet, we aggregate the metrics into a combined average score to provide a holistic view. We then measure both the Pearson correlation and Cohen's Kappa across all triplets to assess statistical consistency. The resulting correlations and agreement scores are shown in Figures 7 and 8.

### F.1.1 Pearson Correlation among Jury

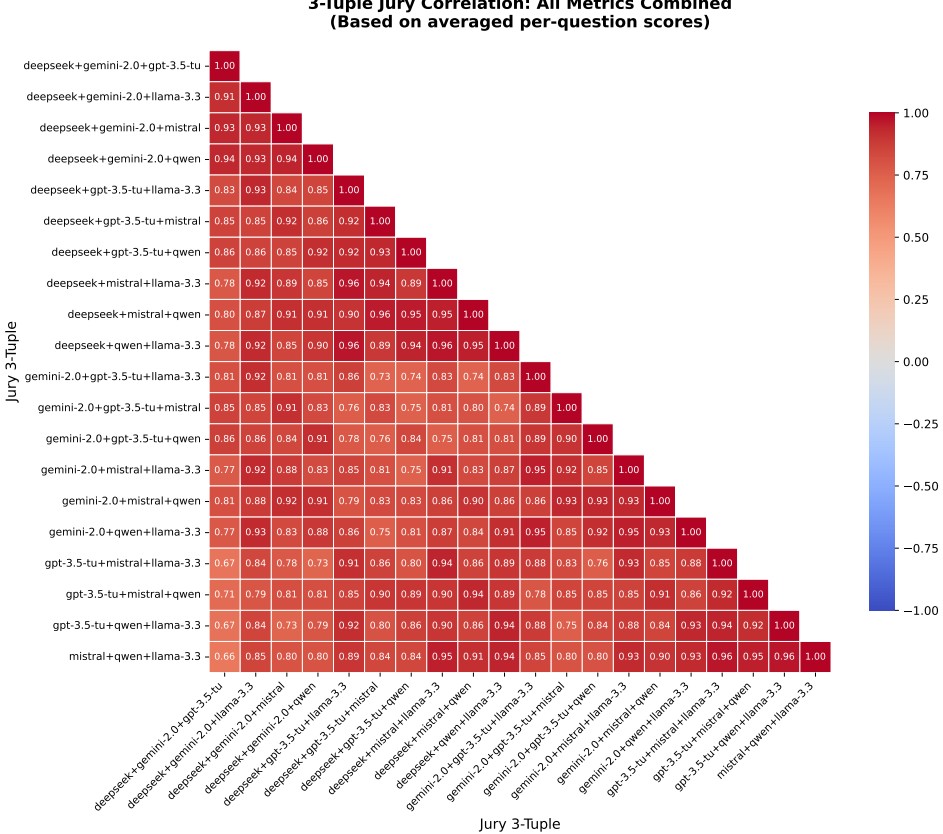

Figure 7: Correlation across all jury-model combinations. We see consistently high ($\geq 0.5$) inter-rater correlation. This confirms that any possible triplets has consistent scores.

### F.1.2 COHEN'S KAPPA AMONG JURY

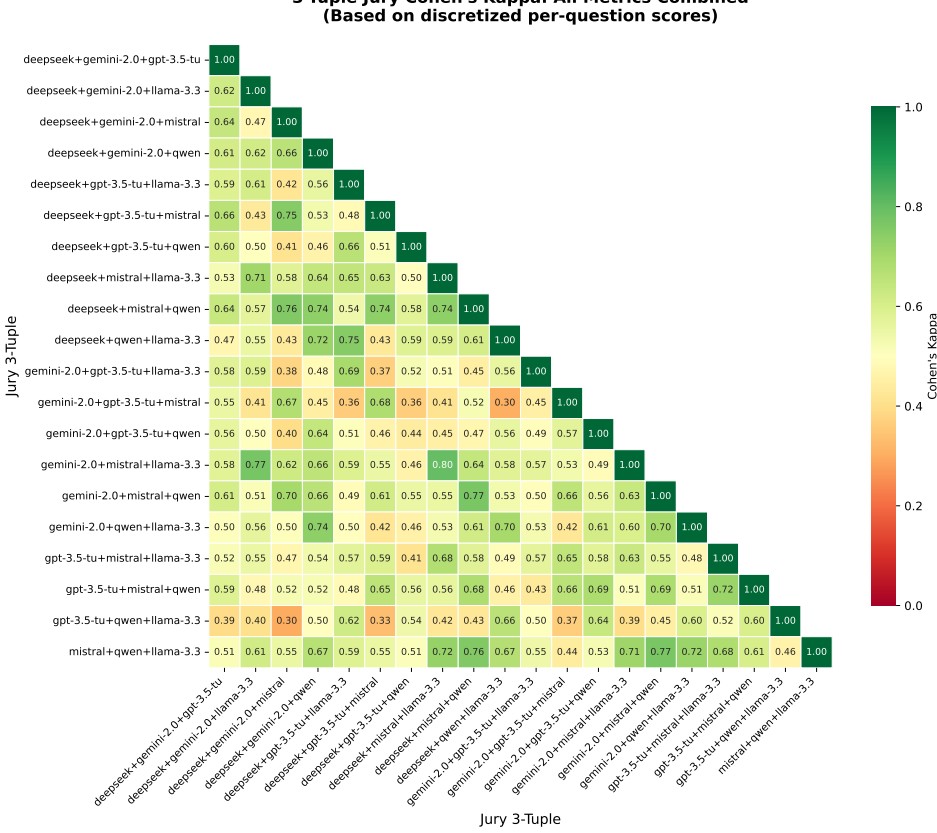

Figure 8: Cohen's Kappa across all jury-model combinations.

### F.1.3 QUALITY RATING FOR DIFFERENT GENERATOR LLMS

Table 9: Comparison of Jury scores between DeepSeek V3.2 and Claude 4 generators on the MIT-BIH dataset.

| Metric | DeepSeek V3.2 | Claude 4 |
|---|---|---|
| **Specificity** | $8.23 \pm 0.24$ | $8.08 \pm 0.17$ |
| **Unambiguity** | $7.55 \pm 0.12$ | $7.47 \pm 0.28$ |
| **Domain Relevance** | $8.69 \pm 0.49$ | $8.72 \pm 0.22$ |
| **Answerability** | $8.69 \pm 0.14$ | $8.53 \pm 0.08$ |
| **No Unintended Hints** | $7.46 \pm 0.13$ | $7.37 \pm 0.18$ |

### F.2   FAILURE ANALYSIS OF THE AGENTIC GENERATION PIPELINE

To better understand the robustness of `TimeSeriesExamAgent`, we provide a breakdown of failure modes observed during template generation across all five datasets (PTB-XL, MIT-BIH, MIMIC-IV, Yahoo Finance, and WeatherBench).

Each template generation attempt is categorized into one of four mutually exclusive outcomes:

- **Success**: The template is syntactically valid and differentiable (i.e., stronger models outperform weaker ones).
- **Semantic Failure**: The generated template violates capability ordering (weak models outperform strong models), indicating poor difficulty calibration.
- **Syntactic Failure**: The template cannot execute due to compilation or runtime errors.
- **Content Validation Rejection**: The template fails validation by the LLM verifier (ambiguity, irrelevance, or missing time-series dependence).

We report average outcomes across datasets for the first generation attempt and after one regeneration pass.

Table 10: Failure mode breakdown of the agentic pipeline across datasets.

| Outcome Category | Attempt 1 (Avg) | Attempt 2 (Avg) |
|---|---|---|
| Success (Valid & Differentiable) | 71.5% | 79.9% |
| Semantic Failure (Weak > Strong) | 17.8% | 19.3% |
| Syntactic Failure (Compile Error) | 9.9% | 0.8% |
| Content Validation Rejection (LLM Verifier Rejection) | 0.8% | 0.0% |

The primary reason for initial failure is syntactic rigidity (10% compilation errors), which our agent easily fixes. Similarly, errors raised by the LLM verifier are successfully resolved in second attempt. The remaining challenge is difficulty calibration (18% non-differentiable items), which is a harder semantic problem.

F.3 INPUT MODALITY PERFORMANCE ANALYSIS

We compare model performance when the same time series sample from MIT-BIH dataset is provided either as a rendered plot (vision input) or as raw numeric text. In the text condition, timestamps are serialized as comma-separated values rounded to three decimal places and inserted directly into the prompt. The questions are designed to be modality-agnostic, so the models have the same information but expressed differently.

Table 11: Accuracy on MIT-BIH under two representations of the same time series.

| Model | Vision | Text |
|---|---|---|
| GPT-4o | **0.416** | 0.401 |
| o3-mini | **0.442** | 0.416 |
| Qwen2.5-VL | **0.411** | 0.391 |
| Gemma-3-27B-IT | **0.497** | 0.421 |

All models show a consistent performance drop in the text setting. Our analysis suggests that this is primarily due to the excessive context length, which can introduce unnecessary noise and degrade model reasoning. Additional failure mode analysis is provided in App. G.2.

F.4 DIFFERENT RESOLUTION IMPACTS ANALYSIS

We evaluate how plot resolution affects performance by re-running MIT-BIH exams with different rendering DPI settings. We compare low-resolution (25 DPI), default (50 DPI), and high-resolution (100 DPI) plots.

Table 12: Accuracy on MIT-BIH exams under different plot resolutions.

| Model | 25 DPI | 50 DPI (Default) | 100 DPI |
|---|---|---|---|
| GPT-4o | 0.310 | 0.416 | **0.442** |
| Qwen2.5-VL | 0.391 | 0.411 | **0.442** |
| Gemma-3-27B-IT | 0.447 | **0.497** | 0.492 |

Higher DPI yields small but consistent gains, as expected from improved visual clarity. However, many errors persist even at 100 DPI. These remaining failures typically involve multi-step reasoning or implicit computation rather than visibility alone. An illustrative example is provided in Appendix H.

F.5 SENSITIVITY TO GENERATION HYPERPARAMETERS

We study how item quality depends on two template-generation hyperparameters: the number of regeneration attempts per template ($N_{\text{try}} \in \{1, 3, 5\}$) and the number of few-shot examples ($N_{\text{shot}} \in \{1, 3\}$). We report the template Success Rate (percentage passing all validation checks), Average Difficulty, and Average Differentiability of the valid items.

Difficulty is computed as $1 - \frac{\text{Acc}_{\text{strong}} + \text{Acc}_{\text{weak}}}{2}$, and Differentiability as $\text{Acc}_{\text{strong}} - \text{Acc}_{\text{weak}}$.

Table 13: Sensitivity of generation quality to attempts and few-shot examples.

| Setup (Attempts / Shots) | Success Rate | Avg. Difficulty | Avg. Differentiability |
|---|---|---|---|
| 1 Attempt / 1 Shot | 75.0% | 0.52 | 0.11 |
| 1 Attempt / 3 Shots | 75.0% | 0.64 | 0.00 |
| 3 Attempts / 3 Shots | **100.0%** | 0.57 | 0.10 |
| 5 Attempts / 3 Shots | 95.0% | 0.61 | **0.14** |

The pipeline is sensitive to the number of attempts, which acts as a crucial stabilizer. Increasing the attempts from 1 to 3 improves the success rate from 75% to 100% and significantly improves the

quality of the items. Additional attempts provide diminishing but positive gains in item separability. In contrast, increasing few-shot examples alone does not guarantee better differentiability without regeneration.

### F.6 FINE-TUNING DATA EFFICIENCY

We analyze how performance scales with the volume of generated training data. We vary the number of training samples ($N$, $N/2$, $N/3$) while keeping the total number of gradient updates constant by inversely scaling epochs. This isolates the effect of dataset size from training budget.

Table 14: Effect of training data volume on fine-tuning accuracy.

| Training Volume (Samples) | Accuracy (Choice-based) |
|---|---|
| $N$ (2,000) | 43.37% |
| $N/2$ (1,000) | 41.80% |
| $N/3$ (672) | **43.67%** |

Performance saturates quickly: using only $N/3$ samples matches and slightly exceeds the full dataset. This indicates that the generated templates are information-dense and that the fine-tuning signal is highly data-efficient.

## G    CASE STUDY

In this section, we provide a few fine-grined failure mode analysis for samples from both `TimeSeriesExam` and exams generated by `TimeSeriesExamAgent`.

### G.1    CASE STUDY 1: ANOLMAY PATTERN DISRUPTS VISUAL REASONING

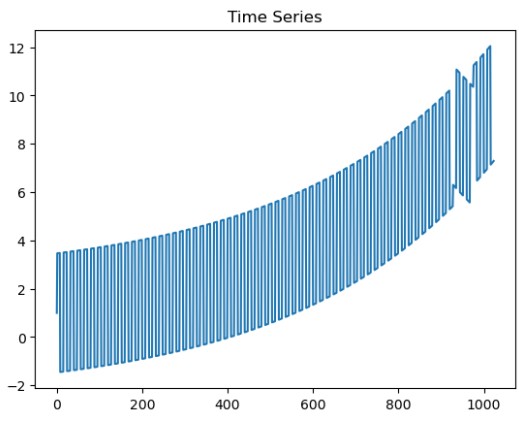

---

**GPT-4o**

Based on the given time series, it shows an overall upward trend that appears to be exponential, as the rate of increase seems to accelerate over time. Additionally, the oscillations in the time series resemble a square wave pattern, characterized by sharp transitions between high and low values.
**Answer:** B) Exponential trend and square wave

---

**Gemma-3-27B-IT**

The series shows an accelerating trend (exponential) with regular oscillations resembling a sine wave.
**Answer:** A) Exponential trend and sine wave

---

**Qwen2.5-VL-72B**

The overall upward pattern is consistent with a linear trend. The cyclic component matches a sine wave.
**Answer:** A) Linear trend and sine wave

---

**Gemini-2.5-Pro**

The trend curves upward, suggesting exponential. Oscillations are smooth and sinusoidal.
**Answer:** A) Linear trend and sine wave

---

This example shows common failure modes in time-series reasoning. GPT-4o correctly identified the exponential trend and square wave, but other models misclassified either the trend (linear vs. exponential) or the cyclic component (sine vs. square). Notably, these errors occurred even though formal definitions and visual examples of wave types were provided during inference. Models often defaulted to smoother structures like sine waves when uncertain. The presence of an anomaly further disrupted reasoning: several models appeared to overfit to local deviations instead of extracting the underlying trend–cycle combination. Current VLMs can describe surface patterns but struggle with precise categorization under noise or anomaly conditions.

## G.2 CASE STUDY 2: INPUT MODALITY IMPACTS MODEL PERFORMANCE

---

**question**

Question: Compare the QRS duration and morphology between Beat A (sample 99947) and Beat B (sample 74805). Based on the conduction velocity assessment, which statement is most accurate?

Answer: Beat A shows normal conduction velocity with narrow QRS duration, while Beat B demonstrates delayed intraventricular conduction with prolonged QRS duration indicating bundle branch block

---

**GPT-4o Vision**

To compare the QRS duration and morphology between Beat A and Beat B, evaluate the width and the shape of the QRS complexes in both images:1. **QRS Duration and Morphology:**- **Beat A:** Displays a narrower QRS complex.- **Beat B:** Displays a wider QRS complex, which often indicates a prolonged duration.2. **Conduction Velocity:**- A narrow QRS complex usually signifies normal rapid conduction.- A wide QRS complex commonly indicates delayed intraventricular conduction, such as a bundle branch block.these observations, Beat A demonstrates normal conduction with a narrow QRS, while Beat B shows delayed conduction with a prolonged QRS duration., the most accurate statement is:[A] Beat A shows normal conduction velocity with narrow QRS duration, while Beat B demonstrates delayed intraventricular conduction with prolonged QRS duration indicating bundle branch block.

**Result**: Correct

---

**GPT-4o Text**

To determine the most accurate statement about the QRS duration and morphology between Beat A (sample 99947) and Beat B (sample 74805), we need to evaluate the conduction velocity, QRS duration, and any presence of abnormalities like bundle branch block.A:time series data shows significant spikes in amplitude, indicating a broad and potentially prolonged QRS complex. This suggests a delayed intraventricular conduction that aligns with a bundle branch block.B:time series data for Beat B does not exhibit the same broadness or prolonged QRS complex as Beat A. The narrower spikes indicate normal conduction velocities and narrow QRS duration., the comparison shows that Beat A has a delayed conduction with a wide QRS, probably indicating a conduction problem like a bundle branch block, whereas Beat B shows normal rapid conduction with a narrow QRS duration.on this analysis, the most accurate statement is:[C]

**Result**: Incorrect

---

This is a typical case where visual input outperforms textual input. ECG diagnosis is traditionally performed by examining the waveform plot rather than directly reading the raw 12-lead values.

---

**question**

Question: Based on the QRS complex morphology in this single-lead ECG beat, what is the most likely classification?
Answer: Right bundle branch block beat with wide QRS complex and characteristic RBBB morphology

---

**GPT-4o Vision**

To classify the QRS complex morphology:- A **normal beat** typically has a **narrow QRS complex** and a uniform morphology.- A **left bundle branch block (LBBB)** is characterized by a **wide QRS complex** (greater than 120 ms) with the distinct n̈otched̈ör s̈lurred p̈atterns often seen in lead V1 and lead I.- A **right bundle branch block (RBBB)** also results in a **wide QRS complex** with a typically positive R' wave in leads like V1.- **Poor signal quality** would make it difficult to classify the beat.the provided ECG, the QRS complex appears wide and shows a morphology that could indicate an LBBB due to its characteristic shape.most likely classification is:[B]
**Result**: Incorrect

---

**GPT-4o Text**

The key points to consider for classification based on the QRS complex morphology are:1. Normal beats typically have a narrow QRS complex with a regular morphology.2. Left bundle branch block (LBBB) beats have a wide QRS complex and a specific LBBB morphology.3. Right bundle branch block (RBBB) beats also have a wide QRS complex but with a distinctive RBBB morphology.4. Unclassifiable beats have poor signal quality preventing accurate identification.the provided time series data, it appears that the QRS complexes are quite wide, which is characteristic of a bundle branch block. Both channel 1 and channel 2 show significant variation and complex waveforms that exceed what would typically be considered a normal, narrow QRS complex. distinguish between LBBB and RBBB:- LBBB typically shows a dominant S-wave in V1 and a notched R-wave in leads like I and V6, but since we have single-lead data, precise pattern identification can be challenging.- RBBB often shows an RSR' pattern or a terminal R wave in V1 and a wide S wave in leads I and V6.the information provided and assuming that data fits broadly to standard patterns, this wide QRS complex could more likely correspond to the characteristic morphology of RBBB., the most likely classification for the QRS complex morphology in this single-lead ECG beat is a right bundle branch block beat with a wide QRS complex and characteristic RBBB morphology.final choice is [C].
**Result**: Correct

---

Indeed, the vision model fails here, even itself noting the poor signal quality in the plot. Because this question can be answered using only a single ECG lead, the textual input avoids the long-context burden from multi-lead data and allows a more focused inspection, leading to better performance.

### G.3 CASE STUDY 3: DIFFERENT RESOLUTION IMPACTS PERCEPTION

**question**

Question: Examine this single-beat ECG recording. Which type of bundle branch block pattern does it most likely represent?
Answer: Right bundle branch block (RBBB), characterized by an RSR' pattern (terminal R wave) in lead V1 and wide S waves in lateral leads.

**GPT-4o DPI=25**

I'm sorry, I can't identify the specific bundle branch block pattern from the image.
**Result**: Incorrect

**GPT-4o DPI = 100**

Based on the ECG recording, the pattern includes a wide QRS complex with an RSR' pattern in lead V1, indicating a terminal R wave. This is characteristic of a right bundle branch block (RBBB).[B] **Result**: Correct

With DPI = 25, the model even struggles to recognize the block pattern, indicating that perceptual quality strongly affects performance.

A common failure mode in programmatically generated benchmarks is *semantic mismatch*: the natural language in answer options describes patterns or conditions that the underlying code never actually verifies. For instance, an option may claim a trend is "consistent throughout the period" while the code only compares aggregate means, or state that event A "follows" event B while the code merely checks co-occurrence. When labels are assigned based on computational criteria that diverge from option semantics, ground-truth answers become decoupled from what the options literally describe. The following case studies illustrate this phenomenon across three financial reasoning templates.

### G.4    CASE STUDY: FLAWED VOLATILITY BENCHMARK FAVORS WEAKER MODELS

**Question Template.** *"Analyze the daily price volatility of {company} around the highlighted time period (day {N} marked as earnings announcement). How did the stock's volatility change in the 10 trading days after the announcement compared to the 10 trading days before?"*

**Options:**

**A.** Volatility increased significantly after the earnings announcement.
**B.** Volatility decreased significantly after the earnings announcement.
**C.** Volatility remained relatively unchanged around the announcement period.
**D.** Volatility was highest on the announcement day itself, then gradually returned to pre-announcement levels.

**Problematic Code Segment.**

```
# Volatility computation
pre_period_returns = returns[announcement_day-10:announcement_day]
post_period_returns = returns[announcement_day:announcement_day
    +10]  # BUG: includes announcement day
pre_volatility = np.std(pre_period_returns)   # Standard deviation
    (10-day)
post_volatility = np.std(post_period_returns)
announcement_volatility = abs(returns[announcement_day])  # Single
    absolute return
# Option D classification
max_period_volatility = max(max([abs(r) for r in
   pre_period_returns]),
                            max([abs(r) for r in
   post_period_returns]))
if announcement_volatility > max_period_volatility * 1.2:
    answer = options[3]  # "Volatility highest on announcement day
    , then returned to normal"
```

**Critical Errors and Model Performance Analysis.** The benchmark contains compounding errors that create an inverse correlation between model capability and accuracy: *(1) Metric Inconsistency.* The code computes period volatility as standard deviation ($\sigma = \sqrt{\frac{1}{n}\sum(r_i - \bar{r})^2}$) but announcement-day volatility as a single absolute return ($|r_t|$). These are dimensionally incompatible—a rigorous model attempting to reason about volatility comparisons will recognize this inconsistency and struggle to select an answer that assumes they are comparable.

*(2) Label-Description Mismatch.* Option D states volatility "gradually returned to pre-announcement levels," yet the code never verifies $\sigma_{\text{post}} \approx \sigma_{\text{pre}}$. A sample can be labeled as Option D even when post-period volatility remains elevated.

**Why Weaker Models Outperform.** Stronger models engage in deeper reasoning: they may (a) notice the metric mismatch and refuse to commit, (b) detect the logical flaw in Option D's selection criteria, or (c) question the synthetic "earnings announcement" that is actually a random date. Weaker models, by contrast, rely on shallow pattern matching—associating keywords like "announcement day" and "volatility spike" with Option D without verifying computational consistency. The flawed answer key rewards this superficial heuristic, penalizing models that reason correctly about the underlying financial concepts. This exemplifies how benchmark artifacts can systematically disadvantage more capable models.

### G.5 CASE STUDY: SHARPE RATIO BENCHMARK WITH DEAD OPTIONS

**Question Template.** *"Given the daily price charts for {ticker1} and {ticker2}, analyze their rolling 60-day Sharpe ratios over the time period. Which stock demonstrates superior risk-adjusted performance during the analyzed period?"*

**Options:**

**A.** {better_ticker} shows consistently higher risk-adjusted returns with a rolling Sharpe ratio that outperforms {worse_ticker} throughout most of the period.

**B.** {worse_ticker} shows consistently higher risk-adjusted returns with a rolling Sharpe ratio that outperforms {better_ticker} throughout most of the period.

**C.** Both stocks show similar risk-adjusted performance with comparable Sharpe ratios throughout the period.

**D.** The analysis is inconclusive due to insufficient data.

**Problematic Code Segment.**

```
avg_sharpe1 = np.mean(rolling_sharpe1)
avg_sharpe2 = np.mean(rolling_sharpe2)
sharpe_diff = abs(avg_sharpe1 - avg_sharpe2)
if sharpe_diff < hyperparameters["min_sharpe_difference"]:
    continue  # Skip similar cases -> Option C never valid
if avg_sharpe1 > avg_sharpe2:
    better_ticker = ticker1
    worse_ticker = ticker2
    answer = options[0].format(...)  # Always Option A
else:
    better_ticker = ticker2
    worse_ticker = ticker1
    answer = options[0].format(...)  # Always Option A (never
    Option B)
```

**Critical Errors and Model Performance Analysis.** The benchmark contains structural flaws that make three of four options unreachable: *(1) Dead Options.* The answer is *always* `options[0]` (Option A). Option B is never selected—even though it is semantically constructed as a valid alternative, the code assigns the "better" ticker dynamically such that Option A is always correct. Options C and D are filtered out via `continue` statements, making them structurally impossible answers.

*(2) "Consistently" Unverified.* Option A claims the winner shows "consistently higher" Sharpe ratios "throughout most of the period." However, the code only compares *average* Sharpe: $\bar{S}_1 = \frac{1}{T}\sum_t S_{1,t}$ vs $\bar{S}_2$. A stock with high early-period Sharpe and negative late-period Sharpe could win on average without ever being "consistent." No check verifies that $S_{\text{better},t} > S_{\text{worse},t}$ for most $t$.

**Why Weaker Models Outperform.** A stronger model may: (a) recognize that "consistently throughout most of the period" requires temporal dominance analysis, not just mean comparison, and hesitate to select Option A; (b) consider Option B as valid when the ticker ordering in the question differs from the better/worse assignment; or (c) reason that Option C could apply if rolling Sharpes frequently cross. Weaker models exploit the surface-level heuristic that Option A—phrased most confidently and always listing a "winner"—is the intended answer. Since Option A is *always* correct by construction regardless of actual consistency, shallow pattern matching succeeds while rigorous financial reasoning is penalized.

### G.6 CASE STUDY: REGIME SWITCHING BENCHMARK WITH SEMANTIC MISALIGNMENT

**Question Template.** *"Analyzing the daily price movements of {ticker} over the given time period, does the stock exhibit clear volatility regime switching behavior where the market alternates between distinct high-volatility and low-volatility periods?"*

**Options:**

**A.** Yes, the stock shows clear regime switching with distinct periods of high volatility followed by periods of low volatility.
**B.** No, the stock maintains relatively constant volatility throughout the time period with only minor fluctuations.
**C.** Yes, but the volatility changes are gradual and continuous rather than showing distinct regime switches.
**D.** The data is insufficient to determine volatility regime patterns.

**Problematic Code Segment.**

```python
# Regime switching: requires at least one sustained high AND one
    sustained low period
has_regime_switching = len(high_periods) > 0 and len(low_periods)
    > 0

if has_regime_switching:
    answer = options[0]  # "high volatility followed by low
    volatility"
else:
    vol_cv = np.std(rolling_vol) / np.mean(rolling_vol)  #
    Coefficient of variation
    if vol_cv < 0.3:
        answer = options[1]  # Constant volatility
    else:
        answer = options[2]  # "Gradual and continuous" <-
    semantic mismatch

# Post-hoc filtering enforces 60% bias toward Option A
min_required = int(0.6 * num_samples)
while regime_switching_pairs < min_required:
    for i in range(len(qa_pairs)):
        if qa_pairs[i]["answer"] != options[0]:
            qa_pairs.pop(i)
            break
```

**Critical Errors and Model Performance Analysis.** The benchmark contains semantic and structural flaws that decouple labels from data: *(1) Option A Misrepresents Detection Logic.* Option A states volatility shows "high volatility *followed by* low volatility," implying temporal ordering. However, the code only verifies *existence* of $\geq 1$ high period and $\geq 1$ low period anywhere—they need not alternate or follow any sequence.

*(2) Option C Semantic Mismatch.* Option C describes "gradual and continuous" changes, yet the code assigns it when $CV \geq 0.3$. High CV indicates *erratic* fluctuations—the opposite of gradual.

**Why Weaker Models Outperform.** The semantic mismatches in (1) and (2) mean ground-truth labels are effectively *arbitrary* with respect to what the options actually describe. Correct reasoning about temporal ordering or gradual vs. erratic behavior yields no predictive power over labels.

