# OpenReview forum: "TimeSeriesExamAgent: Creating Time Series Reasoning Benchmarks at Scale"
_ICLR.cc/2026/Conference — ICLR 2026 Poster_

### Official Review · Reviewer_urhb · 2025-10-26

**Soundness:** 3
**Presentation:** 3
**Contribution:** 2
**Rating:** 4
**Confidence:** 4

**Summary:**

This paper proposes a multi-agent framework in constructing time series related multiple choice questions based on synthetic and real time series. The generator LLM is in charge of creating qa pairs using python functions and verifier LLMs perform validation on generated questions. The paper evaluated four VLMs on the two sets of benchmarks and demonstrated that current LLM cannot perform reliability on relatively more complex questions.

**Strengths:**

The benchmark creation process is almost fully automated except for expert created user instructions.

The paper is easy to follow

The agentic framework can be scaled up.

**Weaknesses:**

Since this is a multi agent framework and is considered to be scaled up in the future, cost is an important consideration that didn't seem to be mentioned or discussed in the paper.

Since the benchmark is fully template based, how to mitigate bias in limited linguistic diversity since humans usually ask questions using diverse expressions and solver LLMs could be sensitive to prompts. And since questions are generated by a single LLM rather than a panel of LLM, how to mitigate bias here that a specific LLM may tend to generate questions in a specific way. I mentioned this because the validation is performed by a panel of LLM.

It feels somewhat like a snake biting its own tail situation where questions are only kept if it's solvable/answerable by an LLM so then why test LLMs on this? And in particular in section 4.3, it's mentioned that you specifically used weaker LLMs as judge, then it didn't make sense to also evaluate answerability of questions.

Although the author discussed this in limitation, this is a relatively important point to bring up is that the evaluation is limited to VLMs. On the more practical side, time series reasoning is hard because you need both numerical computation and reasoning.

The number of evaluated models is very few.

**Questions:**

Are there examples of failed qa pairs? Usually what are reasons that this proposed agentic framework fails? Some analysis on this would be very beneficial.

In sec 4.4, you discussed transferability of the reasoning , but how similar are the synthetic samples to the ECG-QA samples in terms of type of questions asked and formats.

For are questions generated, is it always 4 options? If not, it would be better to include random guessing for table 3 and 4 to know how the models compare to random guessing.

Could you explain a little on why anomaly detection questions are harder than causality questions? Intuitively to verify causal relationships, it requires some statistical analysis, but for anomalies in time series, it's relatively easier to spot given that you are inputting time series as images. If it's unable to recognize that, it may not reflect much on time series reasoning limitation but possible perception limitations -> It's hard to isolate the cause of failures.

---

> ### Author Response · Authors · 2025-11-25
>
> Thank you so much for your valuable comments and time. We are pleased to learn that you believe that our work fills a significant gap in the literature, is clear, detailed, and that our results provide a strong validation signal. We respond to your concerns below.
>
> > W1: Since this is a multi agent framework and is considered to be scaled up in the future, cost is an important consideration that didn't seem to be mentioned or discussed in the paper.
>
> We thank the reviewer for raising this crucial point regarding scalability. We have performed a cost analysis and found that generating a single problem template costs approximately $0.09 using our default setup (default setup provided in Appendix D.5. It is important to note that this is a one-time fixed cost per template. Since a single template can be instantiated into thousands of unique data samples (by varying the underlying time-series data), the amortized cost per evaluation instance becomes negligible (fraction of a cent). Furthermore, our pipeline is optimized for cost-effectiveness by implementing context condensation to minimize token usage during generation when context window is long.
>
> > W2 & W3 regarding LLM-as-a-Judge
>
> Please see the open response to all reviewers on this topic.
>
> > W4: Although the author discussed this in limitation, this is a relatively important point to bring up is that the evaluation is limited to VLMs. On the more practical side, time series reasoning is hard because you need both numerical computation and reasoning.
>
> To address this concern, we generated exams using the MIT-BIH dataset, and evaluated state-of-the-art LLMs by direcly feeding time-series into the context window.  Each timestamp is separated by a comma and round up to 3 decimal places. Our results are as follows:
>
> | Model | MIT-BIH (Vision) | MIT-BIH (Text) |
> | :--- | :---: | :---: |
> | **GPT-4o** | **0.416** | 0.401 |
> | **o3-mini** | **0.442** | 0.416 |
> | **Qwen2.5-VL** | **0.411** | 0.391 |
> | **Gemma-3-27b-it** | **0.497** | 0.421 |
>
> We observe a consistent decrease in model performance when time series are provided in textual form. Our analysis suggests that this is primarily due to the excessive context length, which can introduce unnecessary noise and degrade model reasoning. We analyze failured modes in Appendix H of the revised manuscript.

---

> ### Author Response · Authors · 2025-11-25
>
> > Q1: Are there examples of failed qa pairs? Usually what are reasons that this proposed agentic framework fails? Some analysis on this would be very beneficial.
>
> For actual question answering, we provide a few case study in Appendix H. Our analysis identifies three primary failure modes:
>
>     Compositional Reasoning Breakdowns: Models often default to simple "smooth" priors (e.g., linear trends) and fail to decouple anomalies from the underlying signal structure, even when provided with formal definitions.
>
>     Modality Constraints: Vision models struggle when artifacts or plotting density obscure fine-grained details, while text models fail at holistic comparisons (e.g., beat-to-beat width) due to the burden of processing long numerical contexts.
>
>     Perceptual Granularity: Performance drops significantly at lower resolutions (e.g., 25 DPI), where high-frequency features (like RSR' patterns) become imperceptible, leading to "hallucinated" refusals or errors.
>
> For agentic pipeline, we provide a concrete breakdown of why the framework fails during generation across all five datasets (PTB-XL, MIT-BIH, MIMIC-IV, Yahoo, Weather).
>
> | Outcome Category | Attempt 1 (Avg) | Attempt 2 (Avg) |
> | :--- | :---: | :---: |
> | **Success (Valid & Differentiable)** | 71.5% | 79.9% |
> | **Semantic Failure (Weak > Strong)** | 17.8% | 19.3% |
> | **Syntactic Failure (Compile Error)** | 9.9% | 0.8% |
> | **Jury Failure (LLM Verifier Failure)** | 0.8% | 0.0% |
>
> The primary reason for initial failure is syntactic rigidity (10% compilation errors), which our agent easily fixes. The remaining challenge is difficulty calibration (18% non-differentiable items), which is a harder semantic problem.
>
> > Q2: In sec 4.4, you discussed transferability of the reasoning , but how similar are the synthetic samples to the ECG-QA samples in terms of type of questions asked and formats.
>
> Our generate questions are 4 options multiple choice questions. ECG-QA contains both 4 options MCQAs and some binary options question answering. To better answer your question, we conducted an additional ablation where we fine-tuned the model using out-of-domain data (Finance and Weather) with the same number of gradient steps and evaluated it on the same medical benchmark (ECG-QA). The results reveal a dual impact:
>
>     Question Structure: Fine-tuning on any domain (even irrelevant ones like Finance) raised accuracy to 42.32%, significantly outperforming the Base model (21.8%) and Random guessing (34.9%). This confirms that learning the exam format and validity constraints drives the initial performance jump.
>
>     Domain Alignment: However, fine-tuning on the same domain (PTB-XL → MIMIC-IV) yielded a further gain to 47.0%. This additional ∼5% gap confirms that while structural adaptation is necessary, domain-specific alignment provides a distinct and additive performance boost.
>
> | Method | Accuracy |
> | :--- | :---: |
> | Random answering | 34.9% |
> | Base | 21.8% |
> | Fine-tuned (Other Domain) | 42.32% |
> | **Fine-tuned (Ours)** | **47.0%** |
>
> > Q3: For are questions generated, is it always 4 options? If not, it would be better to include random guessing for table 3 and 4 to know how the models compare to random guessing.
>
> Yes, we always have 4 options for MCQs, we updated the table with random guessing (25%) in the revised manuscript.
>
> > Q4: Could you explain a little on why anomaly detection questions are harder than causality questions? Intuitively to verify causal relationships, it requires some statistical analysis, but for anomalies in time series, it's relatively easier to spot given that you are inputting time series as images. If it's unable to recognize that, it may not reflect much on time series reasoning limitation but possible perception limitations -> It's hard to isolate the cause of failures.
>
> We appreciate this insightful question. While statistical causality is generally complex, we simplified the $\texttt{TimeSeriesExam}$ causality subset to focus strictly on Granger causality with visually distinct features. Conversely, we designed the anomaly detection tasks to require compositional reasoning, knowing that basic visual queries (e.g., "does this series contain a spike?") are often trivial for current vision models. While we acknowledge that completely decoupling perception from reasoning is challenging, this design represents our best attempt to isolate the perceptual factor.

---

> > ### Comment · Reviewer_urhb · 2025-11-27
> >
> > Thank you for the response! I am concerned that the rebuttal indicates the authors modified the benchmark (e.g., simplified the causality subset, adjusted anomaly questions). if question difficulty or structure can be revised post-hoc, it becomes unclear whether model performance differences reflect agent ability or simply reflect the authors’ question-design choices. Benchmarks must remain fixed and reproducible; otherwise, observed results risk being artifacts of template engineering.
> >
> > On the vision model performing better than text model, it makes sense given your question design. But the practicality of this benchmark becomes questionable. Real world time series reasoning tasks predominantly need precise numerical numbers for either analysis or forecasting but I guess this benchmark doesn't cover any. Practically speaking, we won't want VLMs to do the time series analysis but rather a model that can process the numerical information.
> >
> > In the error analysis, what does semantic failure mean? why is weaker model performing better than stronger model problematic? How is weak/strong model determined? Does that mean there is an assumed rank between models before the benchmark creation? that still feels like a recursive setting.

---

> > > ### Author Response · Authors · 2025-12-01
> > > **Author Response Part II (1/2)**
> > >
> > > Dear Reviewer urhb,
> > >
> > > Thank you so much for responding to us!
> > >
> > > > rebuttal indicates the authors modified the benchmark (e.g., simplified the causality subset, adjusted anomaly questions). if question difficulty or structure can be revised post-hoc, it becomes unclear whether model performance differences reflect agent ability or simply reflect the authors’ question-design choices. Benchmarks must remain fixed and reproducible; otherwise, observed results risk being artifacts of template engineering.
> > >
> > > We agree that benchmarks should remain fixed and reproducible. This paper has two aspects: (1) a proof of concept, benchmark called `TimeSeriesExam` and (2) an agent which can create `TimeSeriesExam`-like benchmarks at scale.
> > > We release `TimeSeriesExam` as a benchmark, i.e. it's questions do not change.
> > >
> > > On other hand, `TimeSeriesExamAgent` can create different exams given a dataset (say MIT-BIH). Since the goal of our paper is to evaluate the agent, it is imperative that we run the agent multiple times, resulting in different benchmarks. However, in practice, we expect **domain experts will use `TimeSeriesExamAgent` to create one benchmark**.
> > >
> > > We believe that your confusion might be stemming from this part of our rebuttal:
> > >
> > > > To address this concern, we generated exams using the MIT-BIH dataset, and evaluated state-of-the-art LLMs by directly feeding time-series into the context window.
> > >
> > > In this experiment, the **same exam was used,** albeit with different representations. Moreover results in Section 4.1 (`TimeSeriesExam` Evaluation) are fully reproducible.
> > >
> > > > On the vision model performing better than text model, it makes sense given your question design. But the practicality of this benchmark becomes questionable. Real world time series reasoning tasks predominantly need precise numerical numbers for either analysis or forecasting but I guess this benchmark doesn't cover any.
> > >
> > > As mentioned in the rebuttal and in Appendix H of the revised manuscript, **VLMs perform better than text-only models primarily due to the excessive context length, which can introduce unnecessary noise and degrade model reasoning.** The question templates in `TimeSeriesExam` and `TimeSeriesExamAgent` are not designed to favor any particular kind of model (VLMs, LLMs, or even time series foundation models). Instead, they are designed-based on actual real-world datasets and practical time series reasoning problems.
> > >
> > > Only a subset of time series reasoning tasks require reasoning over precise numerical inputs, e.g. forecasting. Forecasting is a well-studied problem, with several benchmarks such as GIFT-Eval, and are outside the scope of our study, which is limited to time series reasoning problems that domain experts solve on a daily basis. For example, considering how a time series is trending, whether an ECG waveform is indicative of an anomaly, etc. are problem which do not require models to understand numbers with a high degree of precision. These are practical tasks, for which there are few (or no) time series reasoning benchmarks.
> > >
> > > > Practically speaking, we won't want VLMs to do the time series analysis but rather a model that can process the numerical information.
> > >
> > > **Our benchmarks (and the agent) are carefully designed to be practical. Its design is agnostic to the specific models (LLMs, VLMs etc.)** We believe that this is how benchmarks should be designed.
> > >
> > > Note that all models have the same information (i.e. the question and answer options), just expressed differently (images for VLMs, and text for LLMs).

---

> > > > ### Author Response · Authors · 2025-12-01
> > > > **Author Response Part II (2/2)**
> > > >
> > > > > In the error analysis, what does semantic failure mean? why is weaker model performing better than stronger model problematic? How is weak/strong model determined? Does that mean there is an assumed rank between models before the benchmark creation? that still feels like a recursive setting.
> > > >
> > > > These are great questions, thank you so much for asking!
> > > >
> > > > > How is weak/strong model determined? Does that mean there is an assumed rank between models before the benchmark creation?
> > > >
> > > > The strength of a model is determined based on the OpenVLM leaderboard. In our experiments, we select `gpt-4o` as the strong model and `gpt-4o-mini` as the weak model.
> > > >
> > > > > In the error analysis, what does semantic failure mean? why is weaker model performing better than stronger model problematic?
> > > >
> > > > Based on Item Response Theory, a good question should be both **accurate**, and have the ability to differentiate between candidates. It is hard to automatically evaluate whether a question is accurate. Therefore, we **assume that stronger models are likely to perform better than weaker ones, on average**. For some questions, this is not true. We exclude these questions because they are likely to be inaccurate. These questions are said to fail with "semantic error". We have provided a few examples of these questions in the appendix, and have clarified this in the paper.
> > > >
> > > > > that still feels like a recursive setting.
> > > >
> > > > We understand that this assumption might appear limiting. However, we do not believe that this creates a recursive setting because only two models (`gpt-40` and `gpt-4o-mini`) are used to evaluate semantic failures, meanwhile our benchmarks are designed to be evaluated on a wide variety of models. On these resulting benchmarks, we only expect `gpt-4o-mini` to perform worse than `gpt-4o` by design-- this test does not affect the performance or ranking of other models.
> > > >
> > > > We really enjoyed this discussion, and are thankful for your time and comments. We hope that our responses have addressed your concerns. Thank you so much for helping us improve our paper!
> > > >
> > > > Best,
> > > >
> > > > Authors

---

> ### Author Response · Authors · 2025-12-04
> **Additional Experiment on state-of-the-art Model Evaluation**
>
> > W5: The number of evaluated models is very few.
>
> | Model | MIT-BIH | PTB-XL | MIMIC-IV W | YFinance | WeatherBench2 | Average |
> | :--- | :---: | :---: | :---: | :---: | :---: | :---: |
> | random guess | 0.250 | 0.250 | 0.250 | 0.250 | 0.250 | 0.250 |
> | gpt-4o | 0.416 | 0.424 | 0.385 | 0.586 | 0.389 | 0.440 |
> | o3-mini | 0.442 | 0.477 | 0.356 | 0.555 | 0.379 | 0.442 |
> | Qwen2.5-VL-Instruct | 0.411 | 0.490 | **0.439** | 0.572 | 0.368 | 0.456 |
> | Gemma-3-27b-it | 0.497 | **0.517** | 0.370 | 0.534 | 0.232 | 0.430 |
> | gpt-5 | 0.533 | 0.450 | 0.424 | 0.617 | **0.547** | **0.515** |
> | Gemini-2.5-Pro | **0.614** | 0.457 | 0.400 | **0.624** | 0.453 | 0.510 |
>
> We agree that evaluating a broader range of models strengthens the validation of our generated artifacts. To this end, we expanded our evaluation in Table 3 to include frontier models, specifically **GPT-5** and **Gemini-2.5-Pro**. We note that GPT-5 strictly outperforms its predecessor, GPT-4o, across all categories. This improvement confirms that our benchmark effectively distinguishes between model capabilities within the same family. Secondly, while these models demonstrate improved reasoning capabilities, the gains are only incremental. Even the strongest model, GPT-5, achieves only **~51.5% average accuracy**. This reinforces our core argument above: passive, single-modality evaluation is insufficient. Truly solving these tasks requires a intelligent, multi-modal, tool-use capabilities pipeline such as an agentic framework.

---

### Official Review · Reviewer_kZtv · 2025-10-30

**Soundness:** 3
**Presentation:** 2
**Contribution:** 2
**Rating:** 4
**Confidence:** 4

**Summary:**

This paper introduces TimeSeriesExam and TimeSeriesExamAgent, a framework for generating scalable benchmarks to evaluate LLM reasoning over time series data. The authors start from a controlled synthetic setup (TimeSeriesExam) that assesses LLM understanding of five reasoning categories (pattern recognition, noise understanding, similarity, anomaly detection, and causality), and then extend it with an agent-based system that automates benchmark creation from real datasets in finance, healthcare, and meteorology. Experiments show that (i) current VLMs still struggle with time series reasoning, and (ii) the automatically generated benchmarks achieve comparable diversity and quality to human-curated datasets like ECG-QA. The paper further includes a transfer learning study showing modest improvements when fine-tuning on generated data.

**Strengths:**

1. The intersection of time-series analysis and LLM reasoning is rapidly emerging. The paper tackles a genuine gap: the lack of scalable, reasoning-oriented benchmarks for time series.
2. The combination of templated generation, IRT-based refinement, and multi-agent verification (planning–generation–validation) is thoughtfully constructed and clearly explained (see Fig. 2 on p. 5).
3. The paper is very detailed, with algorithmic pseudocode (Algorithm 1), hyperparameter tables, and qualitative examples (e.g., ECG and finance question examples on pp. 21–23). This adds credibility and reproducibility.
4. Evaluation covers multiple domains (medical, financial, weather), and the analysis of generated diversity via embedding and Levenshtein metrics (Table 5) is thorough.

**Weaknesses:**

1. While the engineering effort is strong, the research novelty is limited. The approach largely combines existing components (template-based generation, LLM-as-judge, IRT calibration, multi-agent verification) without a clear new algorithmic or theoretical contribution. The work feels more like a comprehensive system paper than a research breakthrough.
2. The reported results, though broad, are not deeply analyzed. For example:
- Table 4 results (< 55% accuracy) confirm that models perform poorly, but there is little insight into why or what specific reasoning capabilities fail.
- The fine-tuning improvement (21.8 → 47.0%) in Table 7 is promising but lacks ablations—was this due to better question structure, domain alignment, or synthetic diversity?
3. The paper relies heavily on LLM-as-a-judge for quality assessment (G-Eval + model juries). While practical, this is circular—using the same family of models to judge generated questions about themselves risks hidden bias. No human evaluation beyond anecdotal clinician feedback is systematically quantified.
4. Although the agent framework is framed as “scalable,” the actual generation still depends on domain-specific prompts and datasets (discussed in Sec. 5). The system’s generalization to unseen domains is not convincingly demonstrated.
5. The paper is dense and occasionally reads like a project report. The core message is sometimes buried under implementation details. For an ICLR audience, the novelty and research implications should be foregrounded, while extensive technical appendices could be condensed.

**Questions:**

1. How does TimeSeriesExamAgent compare to recent agentic benchmark generators like BenchAgents (2024) in quantitative efficiency (cost, generation speed, diversity per token)?
2. Can the authors provide human evaluation results verifying that LLM-judged “quality” correlates with expert judgment?
3. How does the performance of LLMs differ when reasoning over text-encoded vs. visualized time series inputs (as briefly noted in Sec. 5)?
4. Can the framework generalize to non-numeric structured domains (e.g., tabular, event logs) without major prompt re-engineering?

---

> ### Author Response · Authors · 2025-11-25
>
> Dear Reviewer kZtv,
>
> Thank you so much for your valuable comments and time. We are pleased to learn that you believe that our work fills a significant gap in the literature, is clear, detailed, and that our results provide a strong validation signal. We respond to your concerns below.
>
> > W1: While the engineering effort is strong, the research novelty is limited. The approach largely combines existing components (template-based generation, LLM-as-judge, IRT calibration, multi-agent verification) without a clear new algorithmic or theoretical contribution. The work feels more like a comprehensive system paper than a research breakthrough.
>
> We appreciate the reviewer’s feedback. We acknowledge that constructing a realistic and scalable benchmark with real-world datasets involves significant engineering effort. We believe that the novelty of our research lies in the methodology itself. Specifically, generating (time series) reasoning benchmarks automatically is an understudied problem. Our proposed agentic framework provides a simple but low effort, and effective solution to this challenging problem. Regarding the depth of analysis, we agree that the initial manuscript prioritized describing the pipeline. In the revision, we will shift the balance to include deeper insights into why models fail and how fine-tuning succeeds.
>
> > W2: The reported results, though broad, are not deeply analyzed.
>
> (1) We have expanded Appendix H with detailed case studies analyzing specific failure modes. Our analysis reveals two primary bottlenecks:
>
>     Perceptual Granularity: As evidenced by our ablation on input resolution (DPI) or modality (text vs. vision), the best way to receive data depends on the specific question.
>
>     Compositional Reasoning: Models do not fail not on simple recognition, but on problems that require multi-step reasoning.
>
> (2) We agree that dissecting the source of the fine-tuning improvements is critical. To isolate the effects of structural learning and domain-specific alignment, we conducted an experiment where we fine-tuned the model using out-of-domain data (Finance and Weather) with the same number of gradient steps and evaluated it on the same external benchmark (ECG-QA).
>
> The results reveal a dual impact:
>
>     Question Structure: Fine-tuning on any domain (even irrelevant ones like Finance) raised accuracy to 42.32%, significantly outperforming the Base model (21.8%) and Random guessing (34.9%). This confirms that learning the exam format and validity constraints drives the initial performance jump.
>
>     Domain Alignment: However, fine-tuning on the same domain (PTB-XL → MIMIC-IV) yielded a further gain to 47.0%. This additional ∼5% gap confirms that while structural adaptation is necessary, domain-specific alignment provides a distinct and additive performance boost.
>
> | Method | Accuracy |
> | :--- | :---: |
> | Random answering | 34.9% |
> | Base | 21.8% |
> | Fine-tuned (Other Domain) | 42.32% |
> | **Fine-tuned (Ours)** | **47.0%** |
>
> > W3: LLM-as-a-judge concern
>
> Please see the open response to all reviewers on this topic.
>
> > W4: Although the agent framework is framed as “scalable,” the actual generation still depends on domain-specific prompts and datasets (discussed in Sec. 5). The system’s generalization to unseen domains is not convincingly demonstrated.
>
> Thank you for your thoughtful comments. Regarding domain scalability, we apologize for not communicating our intended meaning clearly. Our use of the term “scalable” does not refer merely to generating large quantities of samples without human supervision. Instead, we view scalability as the ability for domain experts to readily extend and adapt the evaluation pipeline to new proprietary datasets and domain-specific knowledge.
>
> In our framework, the manual effort required for adopting a new domain is limited to: 1) writing a domain-specific prompt, and 2) adapting the dataset loader to our standardized query API. While this is not “scalability” in the sense of fully automated cross-domain generalization, it does provide practical domain scalability: experts can efficiently bring their own data and domain knowledge into the system with minimal overhead. We will revise the manuscript to clarify this distinction and avoid potential misinterpretation.
>
> > W5: The paper is dense and occasionally reads like a project report. The core message is sometimes buried under implementation details. For an ICLR audience, the novelty and research implications should be foregrounded, while extensive technical appendices could be condensed.
>
> We sincerely appreciate this feedback. We acknowledge that the intersection of complex system building and scientific evaluation can lead to high information density. In the revised manuscript, we will streamline the main text to prioritize research implications and novel insights. We will move the granular implementation details to the appendix to ensure the core scientific message is foregrounded for the ICLR audience.

---

> ### Author Response · Authors · 2025-11-25
>
> > Q1: How does TimeSeriesExamAgent compare to recent agentic benchmark generators like BenchAgents (2024) in quantitative efficiency (cost, generation speed, diversity per token)?
>
> Thank you for pointing out this important related work. We note that `BenchAgents` and `TimeSeriesExamAgent` serve fundamentally different purposes: `BenchAgents` generates synthetic datasets, whereas `TimeSeriesExamAgent` generates questions conditioned on user-provided real data. Because the inputs and outputs differ, a direct apples-to-apples comparison in generation efficiency is not straightforward. That said, we agree that reporting cost and efficiency is valuable. We have performed a cost analysis and found that generating a single problem template costs approximately $0.09 using our default setup (default setup provided in Appendix D.5. It is important to note that this is a one-time fixed cost per template. Since a single template can be instantiated into thousands of unique data samples (by varying the underlying time-series data), the amortized cost per evaluation instance becomes negligible (fraction of a cent). Furthermore, our pipeline is optimized for cost-effectiveness by implementing context condensation to minimize token usage during generation when context window is long.
>
> > Q3: How does the performance of LLMs differ when reasoning over text-encoded vs. visualized time series inputs (as briefly noted in Sec. 5)?
>
> To address this concern, we generated exams using the MIT-BIH dataset, and state-of-the-art LLMs by direcly feeding time-series into the context window.  Each timestamp is separated by a comma and round up to 3 decimal places. Our results are as follows:
>
> | Model | MIT-BIH (Vision) | MIT-BIH (Text) |
> | :--- | :---: | :---: |
> | **GPT-4o** | **0.416** | 0.401 |
> | **o3-mini** | **0.442** | 0.416 |
> | **Qwen2.5-VL** | **0.411** | 0.391 |
> | **Gemma-3-27b-it** | **0.497** | 0.421 |
>
> > Q4: Can the framework generalize to non-numeric structured domains (e.g., tabular, event logs) without major prompt re-engineering?
>
> Yes. The framework is designed to operate on structured inputs and does not rely on properties unique to time-series data. As long as the domain provides structured representations (e.g., tabular rows, event logs, categorical features), the question template can be expressed in code-like or schema-based form and processed by our generation LLM without major prompt re-engineering.

---

### Official Review · Reviewer_aEWY · 2025-10-31

**Soundness:** 3
**Presentation:** 3
**Contribution:** 2
**Rating:** 4
**Confidence:** 4

**Summary:**

The paper proposes a two-level approach for evaluating time series reasoning in large language models. The first level is a controlled multiple choice benchmark named TimeSeriesExam that uses synthetic series to probe five core capabilities including pattern recognition, noise understanding, similarity analysis, anomaly detection, and causality. The second level is TimeSeriesExamAgent, an agentic pipeline that takes a natural language task description and a dataset loader, then generates executable Python question templates with rule based answer computation. The pipeline applies three verification stages including structure checks, LLM judging, and capability aligned filtering, with the goal of producing high quality, unambiguous, domain specific items at scale and with limited expert time. The paper evaluates across healthcare, finance, and weather datasets. It reports that modern vision language models still struggle on higher order time series reasoning, and that the automatically generated benchmarks reach diversity comparable to manually curated sets.

**Strengths:**

1. In this paper, authors provide clear articulation of the practical bottleneck. The paper motivates the need for scalable domain specific evaluation rather than one size fits all benchmarks and gives an agentic design that maps cleanly to practitioner workflows.

2. In the paper, generating templates rather than individual questions improves reuse and ensures rule based answer derivation from real series, which supports repeatability and reduces silent errors. The rationale is explicitly stated and connected to implementation details.

3. The combination of syntax checks, LLM judging, and capability aligned filtering is a thoughtful safeguard against ill posed or trivial items and is presented with a concrete architectural sketch.

4. The authors document how clinician feedback on ECG terminology guided prompt refinement, which improves face validity and reduces ambiguity for medical users.

**Weaknesses:**

1. In the paper, contribution hierarchy and granularity are not fully disentangled. The proof of concept and the agentic system are linked, but the paper would benefit from a clearer map of which empirical claims rely on synthetic control versus real data generation. The introduction hints at both goals, but the boundaries remain blurred in places.

2. In this paper, heavy reliance on LLM as a judge raises construct validity concerns. The paper provides judging rubrics, yet it remains unclear how consistent the scores are across different judges and prompts, or how often human experts overrule LLM judgments during verification.

3. The evaluation uses base64 plot images at low DPI. This choice can cap the information bandwidth and may conflate perception limits with reasoning limits. A sensitivity analysis on plot resolution and rendering parameters would strengthen the claim that failures reflect reasoning rather than visibility.

4. The finetuning setup is described with clear hyperparameters, but the paper does not report confidence intervals or significance for the reported improvements, nor ablations on the volume or composition of generated items.

5. The idea that weak models should not outperform strong ones on valid items is intuitive, but the authors does not quantify how many candidate templates are filtered for this reason or provide failure taxonomies that would help others replicate the filtering policy.

**Questions:**

1. How do you guarantee label faithfulness for items derived from real datasets where the answer is computed by a template rather than given by an external ground truth?

2. How sensitive is item difficulty to the specific hyperparameters of the template functions?

3. Finetuning on generated items improves performance on one downstream benchmark. Does this transfer to an external benchmark that was not seen during template creation and that is annotated independently?

---

> ### Author Response · Authors · 2025-11-25
>
> Dear Reviewer aEWY,
>
> Thank you so much for your valuable comments and time. We are pleased to learn that you believe that our work identifies a significant gap, is novel, and that our results provide a strong validation signal. We respond to your concerns below.
>
>
> > W1: In the paper, contribution hierarchy and granularity are not fully disentangled. The proof of concept and the agentic system are linked, but the paper would benefit from a clearer map of which empirical claims rely on synthetic control versus real data generation. The introduction hints at both goals, but the boundaries remain blurred in places.
>
> Thank you so much for this suggestion. We agree, and are working on re-writing parts of the paper to clearly flesh out the contributions.
>
> > W2: In this paper, heavy reliance on LLM as a judge raises construct validity concerns. The paper provides judging rubrics, yet it remains unclear how consistent the scores are across different judges and prompts, or how often human experts overrule LLM judgments during verification.
>
> Please see the open response to all reviewers on this topic.
>
> > W3: The evaluation uses base64 plot images at low DPI. This choice can cap the information bandwidth and may conflate perception limits with reasoning limits. A sensitivity analysis on plot resolution and rendering parameters would strengthen the claim that failures reflect reasoning rather than visibility.
>
> Thank you for your thoughtful comments. To further assess the effect of image clarity on model performance, we re-evaluated the MIT-BIH dataset using DPI = 25 and DPI = 100 (o3-mini recently disabled the vision component so we left them as blank).
>
> | Model | MIT-BIH (25 DPI) | MIT-BIH (50 DPI, Default) | MIT-BIH (100 DPI) |
> | :--- | :---: | :---: | :---: |
> | **GPT-4o** | 0.310 | 0.416 | **0.442** |
> | **o3-mini** | - | **0.442** | - |
> | **Qwen2.5-VL** | 0.391 | 0.411 | **0.442** |
> | **Gemma-3-27b-it** | 0.447 | **0.497** | 0.492 |
>
> Overall, we observe that increasing DPI leads to small but consistent performance gains, which is expected given the improved visual clarity. However, several questions remain incorrect even at higher resolutions. These cases often require non-trivial computation or reasoning steps that current visual-language models struggle to handle. We include an illustrative example of such a failure case in Appendix H of the revised manuscript.
>
> > W4: The idea that weak models should not outperform strong ones on valid items is intuitive, but the authors does not quantify how many candidate templates are filtered for this reason or provide failure taxonomies that would help others replicate the filtering policy.
>
> For agentic pipeline, we provide a concrete breakdown of why the framework fails during generation across all five datasets (PTB-XL, MIT-BIH, MIMIC-IV, Yahoo, Weather).
>
> | Outcome Category | Attempt 1 (Avg) | Attempt 2 (Avg) |
> | :--- | :---: | :---: |
> | **Success (Valid & Differentiable)** | 71.5% | 79.9% |
> | **Semantic Failure (Weak > Strong)** | 17.8% | 19.3% |
> | **Syntactic Failure (Compile Error)** | 9.9% | 0.8% |
> | **Jury Failure (LLM Verifier Failure)** | 0.8% | 0.0% |
>
> The primary reason for initial failure is syntactic rigidity (10% compilation errors), which our agent easily fixes. The remaining challenge is difficulty calibration (18% items which cannot be differented), which is a harder semantic problem. We will revise the manuscript to reflect this change.
>
> > W5: The finetuning setup is described with clear hyperparameters, but the paper does not report confidence intervals or significance for the reported improvements, nor ablations on the volume or composition of generated items.
>
> We appreciate the reviewer's suggestion to rigorously evaluate the impact of fine-tuning data volume. To address this, we conducted an ablation study varying the number of generated training samples (N, N/2, and N/3) while keeping the total number of gradient updates constant (by inversely scaling the number of epochs). As shown in the table below, we observe that performance is highly data-efficient. The model achieves its peak performance (~43.7%) with as few as ~672 samples trained for 3 epochs, performing comparably to (and slightly better than) the full dataset of 2,000 samples trained for 1 epoch. This saturation suggests that the generated templates are information-dense.
>
> | Training Volume (Samples) |  Accuracy (Choice-based)|
> | :--- | :---: |
> | **$N$ (2,000)** | 43.37% |
> | **$N/2$ (1,000)** | 41.80% |
> | **$N/3$ (672)** | **43.67%** |

---

> ### Author Response · Authors · 2025-11-25
>
> > Q1: How do you guarantee label faithfulness for items derived from real datasets where the answer is computed by a template rather than given by an external ground truth?
>
> Do you mind clarifying what do you mean by label faithfulness in this context?
>
> > Q2: How sensitive is item difficulty to the specific hyperparameters of the template functions?
>
> We conducted an ablation study varying two key hyperparameters: the number of generation attempts per template ($N_{try} \in \{1,3,5\}$) and the number of few-shot templates ($N_{shot} \in \{1,3\}$) provided to the generator. We measured the Success Rate (percentage of templates passing all checks) and the resulting Differentiability of the valid items.
>
> As shown in the table below, the pipeline is sensitive to the number of attempts, which acts as a crucial stabilizer. Increasing the attempts from 1 to 3 improves the success rate from 75% to 100% and significantly improves the quality of the items. Notably, with only 1 attempt, our agent struggled to generate differentiable items (Avg Differentiability ≈ 0.00).
>
> | Setup (Attempts / Shots) | Success Rate | Avg. Difficulty | Avg. Differentiability |
> | :--- | :---: | :---: | :---: |
> | **1 Attempt / 1 Shot** | 75.0% | 0.52 | 0.11 |
> | **1 Attempt / 3 Shots** | 75.0% | 0.64 | 0.00 |
> | **3 Attempts / 3 Shots** | **100.0%** | 0.57 | 0.10 |
> | **5 Attempts / 3 Shots** | 95.0% | 0.61 | **0.14** |
>
> *Note: Difficulty is calculated as $1 - \frac{Acc_{strong} + Acc_{weak}}{2}$, and Differentiability as $Acc_{strong} - Acc_{weak}$.*
>
> > Q3: Finetuning on generated items improves performance on one downstream benchmark. Does this transfer to an external benchmark that was not seen during template creation and that is annotated independently?
>
> We appreciate this suggestion regarding generalization. We wish to highlight that Table 7 already addresses this via our evaluation on ECG-QA [1]. This benchmark was annotated independently of our framework. Crucially, we ensured a strict separation of data sources: our model was fine-tuned on generated data from PTB-XL, but evaluated on the MIMIC-IV split of ECG-QA. This confirms that the model is not merely memorizing instances, but successfully transferring reasoning capabilities to unseen data from a different source within the same domain.
>
> [1] Oh, Jungwoo, et al. "Ecg-qa: A comprehensive question answering dataset combined with electrocardiogram." Advances in Neural Information Processing Systems 36 (2023): 66277-66288.

---

### Official Review · Reviewer_RbCS · 2025-11-01

**Soundness:** 3
**Presentation:** 3
**Contribution:** 2
**Rating:** 6
**Confidence:** 4

**Summary:**

This paper proposes a new framework for generating time series reasoning benchmarks at scale, arguing that existing benchmarks are manually curated, narrow, and expensive to create. The authors' contribution is twofold:

TimeSeriesExam: A "proof-of-concept" benchmark consisting of multiple-choice questions based on synthetic time series data. It aims to test five core, domain-agnostic reasoning skills: pattern recognition, noise understanding, similarity, anomaly detection, and causality.

TimeSeriesExamAgent: A multi-agent framework designed to automatically generate new, domain-specific benchmarks from real-world datasets. This agentic pipeline (using Generator, Concept, Verifier, and Student LLMs) takes a dataset (e.g., PTB-XL) and a prompt, and outputs a new "exam" with questions derived from that data.

The paper's central findings are that (1) its agent can generate benchmarks with diversity "comparable to human-curated" ones, and (2) all state-of-the-art LLMs (GPT-4o, Gemini 2.5-Pro) perform poorly on both the synthetic and the agent-generated exams, suggesting their time series reasoning capabilities are "limited."

**Strengths:**

Valid and Important Problem: The paper correctly identifies a significant gap in the field. Assessing the true reasoning of LLMs on time series data (beyond simple forecasting accuracy) is a critical and under-explored problem. The goal of creating a scalable benchmark generation process is highly valuable.

Novel Agentic Framework: The core idea of TimeSeriesExamAgent is the paper's strongest contribution. Using a multi-agent pipeline to generate, verify, and filter questions is a novel and powerful approach to benchmark creation that is far more scalable than the manual curation.

Sophisticated Verification Loop: The verification process (Fig. 2) is well-considered. It includes not just a structural check and an "LLM Verifier," but also a "Capability-Aligned Filtering" step that uses "Student LLMs." The idea of discarding questions that weaker models perform better on is a clever and sound method for filtering out flawed or noisy questions.

Strong Validation Signal (Table 7): The transfer learning experiment is a key piece of evidence. Showing that fine-tuning a VLM on the agent-generated data improves its performance on a separate, human-curated dataset (ECG-QA) strongly suggests that the generated questions provide a meaningful and useful learning signal.

**Weaknesses:**

1. Recursive and Subjective Evaluation:

The paper's quality control relies heavily on an "LLM-as-a-judge" and "LLM-as-a-Jury" (Table 6, Appendix G.1) to assess the quality of the generated questions (e.g., "Specificity," "Unambiguity"). This creates a recursive, self-referential loop: an LLM (Generator) creates a question, which is then validated by another LLM (Verifier), to create a benchmark that is then used to... test LLMs.

This "LLM-evaluates-LLM" methodology is notoriously subjective, prone to bias (e.g., favoring its own "style"), and lacks the objective rigor required for a foundational benchmark. While the authors present this as a feature, it's a methodological weakness that introduces significant unreliability.

2. Incomplete Model Comparison:

While the paper tests several SOTA VLMs, the evaluation is incomplete. The primary TimeSeriesExamAgent framework is designed to generate text and time series questions from real-world data, yet the evaluation is performed only by VLM-style models that take images (Section H). The paper even notes this limitation itself (Section 5, "Limited Evaluation Mode"), stating that "certain question types... are particularly difficult to answer from images alone." This mismatch between the generated data (which is numerical) and the evaluation mode (which is visual) is a strange and limiting choice. The framework is not evaluated against text-only LLMs coupled with numerical data, which would be a more direct test of reasoning over the raw time series.

**Questions:**

See weakness

---

> ### Author Response · Authors · 2025-11-25
>
> Thank you so much for your valuable comments and time. We are pleased to learn that you believe that our work identifies a significant gap, is novel, and that our results provide a strong validation signal. We respond to your concerns below.
>
> > W1: The paper's quality control relies heavily on an "LLM-as-a-judge" and "LLM-as-a-Jury" (Table 6, Appendix G.1) to assess the quality of the generated questions (e.g., "Specificity," "Unambiguity"). This creates a recursive, self-referential loop: an LLM (Generator) creates a question, which is then validated by another LLM (Verifier), to create a benchmark that is then used to... test LLMs. This "LLM-evaluates-LLM" methodology is notoriously subjective, prone to bias (e.g., favoring its own "style"), and lacks the objective rigor required for a foundational benchmark. While the authors present this as a feature, it's a methodological weakness that introduces significant unreliability.
>
> Please see the open response to all reviewers on this topic.
>
> > W2: While the paper tests several SOTA VLMs, the evaluation is incomplete. The primary TimeSeriesExamAgent framework is designed to generate text and time series questions from real-world data, yet the evaluation is performed only by VLM-style models that take images (Section H). The paper even notes this limitation itself (Section 5, "Limited Evaluation Mode"), stating that "certain question types... are particularly difficult to answer from images alone." This mismatch between the generated data (which is numerical) and the evaluation mode (which is visual) is a strange and limiting choice. The framework is not evaluated against text-only LLMs coupled with numerical data, which would be a more direct test of reasoning over the raw time series.
>
> To address this concern, we generated exams using the MIT-BIH dataset, and evaluated state-of-the-art LLMs by direcly feeding time-series into the context window.  Each timestamp is separated by a comma and round up to 3 decimal places. Our results are as follows:
>
> | Model | MIT-BIH (Vision) | MIT-BIH (Text) |
> | :--- | :---: | :---: |
> | **GPT-4o** | **0.416** | 0.401 |
> | **o3-mini** | **0.442** | 0.416 |
> | **Qwen2.5-VL** | **0.411** | 0.391 |
> | **Gemma-3-27b-it** | **0.497** | 0.421 |
>
>
> We observe a consistent decrease in model performance when time series are provided in textual form. Our analysis suggests that this is primarily due to the excessive context length, which can introduce unnecessary noise and degrade model reasoning. We analyze failured modes in Appendix H of the revised manuscript.

---

### Author Response · Authors · 2025-11-25
**Concern on LLM-as-a-Judge for quality control**

Thank you all for raising this important methodological concern. We agree that benchmarks built for LLMs, by LLMs, and evaluated using LLMs should be designed with care. In our work, we only leverage LLMs' ability to generate coherent text and evaluate simple linguistic properties.

1. **LLM Verifier Filters Trivially Invalid Items:** As shown in Appendix D, the verifier is not checking whether a question is correct or solvable. Instead, it performs simple linguistic filtering that removes trivially invalid items such as:

```
Domain Relevance: “Is the question related to {exam_type} time series analysis?” (e.g., avoid asking ECG questions for a weather dataset).

Time-Series Dependence: “Can the question be answered without looking at the time series?” This removes tautological or content-free questions.
```

These are simple textual checks that are closer to grammar where even modest LLMs have been shown to perform reliably [1,2]. None of these checks assess content correctness.

2. **A Jury of LLMs to Measure Quality:** In Table 6, we compare our generated questions against established domain benchmarks (ECG-QA, FinMMD) again on linguistic qualities such as ambiguity and domain relevance. These criteria do not evaluate correctness or difficulty. The purpose is simply to show that the generated questions are linguistically similar to existing benchmarks.

To reduce bias, we employ a _jury_ of diverse LLMs (comprising of both open-source and closed-source models). To further demonstrate that our results are unbiased, we conducted 2 experiments: in the first experiment we kept the generator LLM fixed, and evaluted the exams with different LLM juries. In the second experiment, we kept the LLM jury fixed, and generated exams using different LLM generators. In both experiments, we see that our conclusions do not change significantly. Detailed results can be found in Appendix I of the revised manuscript.

**Experiment 1: Fixed Generator, Different Juries**
Exams were generated using `Claude-4`, and evaluated using 3-model juries drawn from a fixed pool of LLMs (`Gemini-2.0`, `Deepseek-V3.2`, `GPT-3.5 Turbo`, `Qwen-2.5-VL`, `LLama-3.3`). We observed that scores from most juries were moderately to highly correlated (Cohen's $\kappa \geq 0.5$).

**Experiment 2: Fixed Jury, Different Generators**
We fixed the jury to the default configuration (`Gemini-2.0`, `GPT-3.5-Turbo`, and `Qwen2.5-VL-72B-Instruct`; Gemini-1.5-Pro was discontinued so we substitute with 2.0), and compared two different generator LLMs: `Claude 4` and `DeepSeek V3.2`. We specifically picked DeepSeek because it is disjoint from both the Jury and Verifier model families (Qwen, Gemini, GPT), ensuring strict independence. Unlike the previous experiment, changing the generator alters the specific questions produced. Therefore, we fixed the underlying data source to **MIT-BIH** and evaluated the statistical distribution (mean $\pm$ standard deviation) of the jury scores

| Metric | DeepSeek V3.2 | Claude 4 |
| :--- | :---: | :---: |
| **Specificity** | 8.23 $\pm$ 0.24 | 8.08 $\pm$ 0.17 |
| **Unambiguity** | 7.55 $\pm$ 0.12 | 7.47 $\pm$ 0.28 |
| **Domain Relevance** | 8.69 $\pm$ 0.49 | 8.72 $\pm$ 0.22 |
| **Answerability** | 8.69 $\pm$ 0.14 | 8.53 $\pm$ 0.08 |
| **No Unintended Hints** | 7.46 $\pm$ 0.13 | 7.37 $\pm$ 0.18 |

While DeepSeek V3.2 exhibits slightly higher raw means, **the results are statistically comparable, with the scores of both models falling within one standard deviation of each other across all categories.** This confirms that our generation pipeline is robust to the choice of state-of-the-art generator LLM.

3. **Our Benchmarks are Not Circular:** The benchmark produced by `TimeSeriesExamAgent` and its difficulty does not come from generator LLM's internal knowledge. Instead, it arises from:

**External data + domain knowledge**
Each question requires interpreting time-series data and applying domain concepts (e.g., ECG morphology, financial event patterns). An LLM cannot answer these by relying on memorized knowledge or stylistic priors. It must reason over the data instance.

**Localized IRT via “LLM student verifiers”**
We use relative performance across models of varying abilties to identify questions that are difficult or discriminative. This is analogous to localized Item Response Theory. Invalid and overly easy questions are eliminated as a part of this process.

Together, these components ensure that the evaluation is not testing models on questions tailored to LLM preferences. Instead, it tests whether models can integrate external data with domain knowledge.

[1] Zheng, Lianmin, et al. "Judging llm-as-a-judge with mt-bench and chatbot arena." Advances in neural information processing systems 36 (2023): 46595-46623.

[2] Gilardi, Fabrizio, Meysam Alizadeh, and Maël Kubli. "ChatGPT outperforms crowd workers for text-annotation tasks." Proceedings of the National Academy of Sciences 120.30 (2023): e2305016120.

---

### Author Response · Authors · 2025-12-01
**General Response (1/2)**

Dear Reviewers and Area Chair(s),

Thank you so much for your time time and effort dedicated to reviewing our work. We are encouraged by the positive reception and summarize the key merits highlighted by the reviewers below:

1.  **Addresses a Critical Gap:** Reviewers noted that our work targets an important and relevant problem in the literature, to address the lack of **scalable time series reasoning benchmarks** [RbCS, aEWY, kZtv].
2.  **Intuitive & Practical Verification Loop:** Reviewers found our multi-stage verification process including IRT-based refinement, multi-agent verifications etc. to ensure robust and automated exam generation, to be thoughtful, well-motivated, important, and well-designed [RbCS, aEWY, kZtv, urhb].
3.  **Robustness via Templated Generation:** The choice to generate algorithmic templates rather than static Q&A samples was recognized as an interesting and key design choice that enhances the framework's robustness and scalability [RbCS, aEWY, kZtv].
4.  **Extensible Multi-Domain Evaluation:** Reviewers acknowledged the framework's adaptability across diverse datasets. By minimizing manual effort to just expert prompting and data loading [urhb, kZtv], our approach efficiently empowers experts to evaluate models on proprietary or specialized data.

We wish to emphasize that this work goes beyond a static benchmark; we provide a **generative tool** that creates customized evaluations based on user-specific prompts and proprietary datasets. As generic benchmarks become insufficient for specialized fields (e.g., cardiology), our approach represents a step towards democratizing evaluation, allowing domain experts to rigorously test evolving model architectures on their own data.

---

> ### Author Response · Authors · 2025-12-03
> **General Response (2/2)**
>
> At the same time, we received valuable feedback from our reviewers. Your feedback highlighted critical areas for refinement, and we have conducted a comprehensive suite of new experiments to address these concerns during the rebuttal phase.
>
> 1. **Validity of LLM-as-a-judge:** We acknowledge concerns about potential circularity when using LLMs to evaluate other LLMs [RbCS, aEWY, kZtv, urhb] To address this valid critique, we reinforced our methodology with strict safeguards. We reiterate that our verifiers perform **shallow linguistic checks** (e.g., domain relevance) rather than deep solution verification, which minimizes the risk of model-specific biases propagating. Second, we use disjoint model families (Claude as generator, GPT as verifier) to prevent same-family optimization artifacts. Third, our Jury stability analysis demonstrates that results remain consistent across different judge combinations, indicating robust evaluation rather than judge-specific quirks. Together, these design choices enable scalable evaluation while maintaining benchmark objectivity.
>
> 2. **Multi-modality & Perception Limits:** We conducted an ablation on the MIT-BIH dataset comparing vision vs. text-only modes. Text-only performance degraded significantly (confirming that visual cues are necessary for physiological signals), but crucially, both modalities achieved < 55% accuracy. This low performance ceiling strongly validates our core motivation: passive perception alone—regardless of modality—is insufficient for the questions generated by `TimeSeriesExamAgent`. High performance on our benchmarks requires genuine agentic capabilities: dynamic modality selection, computational tool use (code execution), and multi-step reasoning. The benchmark successfully distinguishes between systems that merely perceive and those that can actively reason and problem-solve.
>
> 3. **Rigorous Pipeline Analysis:** We agree with the reviewers that the robustness of the generation pipeline required deeper quantitative evidence [aEWY, urhb]. In response, we have added extensive ablation studies to the revision, including a **taxonomy of failure modes** (differentiating syntactic compilation errors from semantic filtering), a sensitivity analysis of **generation hyperparameters** (attempts and shots), and a **fine-tuning data efficiency** study.
>
> 3. **Rigorous Pipeline Analysis:** We agree that the generation pipeline required deeper quantitative validation [aEWY, urhb]. In response, we have added extensive ablation studies: (1) a **failure mode taxonomy** categorizing syntactic errors vs. semantic issues, (2) **hyperparameter sensitivity analysis** examining the impact of generation attempts and few-shot examples on question quality, and (3) a fine-tuning data efficiency study quantifying performance gains across different generated training set sizes.
>
> Changes to our paper made as a result of your feedback are shown in deep blue.
>
> We believe these additions meaningfully advance the quality and clarity of the work. The reviewers’ critiques prompted several substantial improvements—expanded analyses, multimodal ablations, and rigorous pipeline validation—that directly resolve the concerns raised. In addressing these points, the paper has become both stronger and more comprehensive: components once viewed as limitations are now supported by clear empirical evidence and explicit design justifications. We are confident that the revised manuscript presents a robust, scalable, and impactful contribution to the study of agentic evaluation for time-series reasoning.

---

### Meta-Review · Area_Chair_n6UK · 2026-01-09

**Summary:**

The paper presents TimeSeriesExamAgent, a framework for generating scalable benchmarks to evaluate Time Series reasoning in Large Language Models (LLMs). The work consists of two main components:

(1) TimeSeriesExam: A proof-of-concept benchmark using synthetic data to probe core reasoning skills (e.g., causality, anomaly detection).

(2) TimeSeriesExamAgent: An agentic framework that generates template-based questions from real-world datasets (Finance, Healthcare, Weather) using a multi-step pipeline (Generator, Verifier, Student LLMs).

The authors demonstrate that current State-of-the-Art (SOTA) models (including GPT-4o and Gemini) struggle with these reasoning tasks. Furthermore, they show that fine-tuning models on their agent-generated data improves performance on external human-curated benchmarks (ECG-QA), suggesting the generated data possesses high quality and utility.

**Reviewer Concerns:**

The main concerns are:

(1) Evaluation Modality (Vision vs. Text): Reviewers RbCS, aEWY, and urhb criticized the evaluation for relying solely on Vision-Language Models (VLMs) viewing images, arguing that real-world time series analysis often involves numerical data.

(2) LLM-as-a-Judge Circularity: Multiple reviewers (RbCS, aEWY, kZtv, urhb) worried about the problem of using LLMs to verify questions intended to test LLMs.

(3) Pipeline Robustness & Ablations: Reviewers aEWY and urhb requested more analysis on why generation fails.

(4) Novelty vs. Engineering: Reviewer kZtv noted the work feels like a "comprehensive system paper" rather than a research breakthrough, as it combines existing techniques (CoT, LLM-as-judge, IRT) without a new theoretical contribution. This remains a fundamental property of the paper.

**Reviewer Scores:**

There is a positive score, and 3 scores are marginally below the acceptance threshold. During the rebuttal, most of the concerns were well addressed.

---

### Decision · Program_Chairs · 2026-01-26

Accept (Poster)